# Inferentially-Private Private Information

## ABSTRACT

Information disclosure can compromise privacy when revealed information is correlated with private information. We consider the notion of *inferential privacy*, which measures privacy leakage by bounding the inferential power a Bayesian adversary can gain by observing a released signal. Our goal is to devise an *inferentially-private private information structure* that maximizes the informativeness of the released signal, following the Blackwell ordering principle, while adhering to inferential privacy constraints. To achieve this, we devise an efficient release mechanism that achieves the inferentially-private Blackwell optimal private information structure for the setting where the private information is binary. Additionally, we propose a programming approach to compute the optimal structure for general cases given the utility function. The design of our mechanisms builds on our geometric characterization of the Blackwell-optimal disclosure mechanisms under privacy constraints, which may be of independent interest.

**ACM Reference Format:**
. 2024. Inferentially-Private Private Information. In *Proceedings of ACM Conference (Conference'17)*. ACM, New York, NY, USA, 16 pages. https://doi.org/10.1145/nnnnnnn.nnnnnnn

## 1 INTRODUCTION

Information disclosure is essential to enabling cooperation between entities. However, the information an agent reveals can be correlated with private information that should not be revealed. For example, publicly-traded companies are required to release quarterly earnings reports. These earnings reports are closely correlated to the company's business strategy, which may be considered private. Hence, businesses may be incentivized to tailor the information in their earnings report to maximize the signal (the state of their overall finances) while minimizing disclosure about private information (their business strategy) [11]. More generally, privacy concerns stemming from information disclosure are a pervasive problem that inhibits data sharing and disclosure [20].

In such scenarios of information disclosure, it is natural to ask how much information an observer can infer about private information. *Inferential privacy* captures precisely this notion [8, 12]. Roughly, inferential privacy requires that the adversary's posterior over the secret values is within some bounded ratio of the adversary's prior (a formal definition is provided in §2). Hence natural questions include: *how should one release information subject to an inferential privacy guarantee? What mechanism should one use? Can we find mechanisms that release some state subject to an inferential*

*privacy constraint, while also maximizing the utility of a downstream decision maker who sees only the released information?*

We study these questions under the following setting. Consider a random variable $Y \in \{0, 1\}$, also called the *state*, which represents the information we want to release (e.g., whether quarterly earnings are good or bad). We also consider a *sensitive* random variable (or secret) $S \in \mathcal{S}$, where $\mathcal{S}$ is a finite set. In our earlier example, $S$ might represent the company's business strategies, which should remain private. The state $Y$ and the private information $S$ are correlated: they are jointly drawn from a distribution $\mathbb{P}(S, Y)$, which is assumed to be known to both the data holder (e.g., the company) and the observer (e.g., viewers of the earnings report). We aim to design an information disclosure mechanism that releases an *output signal* random variable $T \in \mathcal{T}$ for some (possibly infinite) set $\mathcal{T}$ of output signals. $T$ should be informative about $Y$, without revealing too much information about $S$. More precisely, we assume there is a reward function $r$ over the state $Y$ and the decision maker's corresponding action $A$, and our utility $u$ over the output signal $T$ is defined as the expected reward maximized over the decision maker's actions, under a certain output signal. Hence, our goal is to design an information structure (which corresponds to a joint distribution $\mathbb{P}(S, Y, T)$) that maximizes expected utility over output signal $T$, subject to an inferential privacy constraint.

Recently, He et al. [14] studied a special case of this problem under a "perfect" inferential privacy constraint (0-inferential privacy as in Definition 2.1); that is, they constrain their information structure to not leak *any* information about $S$. In other words, the observer's posterior over the secret $S$ should be the same as their prior. Their privacy constraint can be viewed as a special, extreme case of inferential privacy. He et al. [14] demonstrate a closed-form information structure that simultaneously achieves perfect privacy and Blackwell-optimality (§2), which is also proved to achieve maximal utility. Their information structure is optimal in the sense that no information structure can be more informative without revealing information about the sensitive information $S$.

In this work, we generalize the formulation of He et al. [14] to accommodate the more general inferential privacy constraint, rather than requiring perfect privacy. In particular, the parametric definition of inferential privacy enables us to explore a broader spectrum of privacy-utility trade-offs. As the inferential power of the adversary varies, so does the utility for the decision maker. We find instances where the decision maker's utility significantly increases by merely loosening the perfect privacy constraint to a stringent inferential privacy level. Specifically, we demonstrate that at any given (nonzero) level of inferential privacy, there is an instance where the difference in utility between mechanisms ensuring perfect privacy and those optimizing utility under the inferential privacy constraint can be arbitrarily large (Prop. 5.1). Note that under our formulation, inferential privacy can be viewed as a special case of other recent formulations for private information disclosure [24]. We discuss our choice of privacy metric in §2.

Our main results are:

(1) **Geometric characterization of Blackwell-optimal solutions:** We provide a geometric characterization of Blackwell-optimal information structures subject to the inferential privacy constraint. A direct implication of our characterization is a bound on the number of signals required to satisfy the privacy constraint. While in principle, the number of possible output signals (i.e., the cardinality of $\mathcal{T}$) required by the optimal structure can be unbounded, we show that it suffices to have at most $|\mathcal{T}| = 3|\mathcal{S}| + 1$ possible signals to achieve Blackwell optimality. As a result, a Blackwell-optimal, inferentially private information structure is exactly computable; when the number of possible secrets is constant, it is computable in polynomial time. Our geometric characterization involves tiling a two-dimensional space with $|\mathcal{S}| \cdot |\mathcal{T}|$ cells, each of which has its width and length and is associated with a positive state (i.e., the true state has value $Y = 1$) or a negative one (i.e., $Y = 0$). A particular tiling of these cells fully determines the joint distribution over the state, secret, and output signal $\mathbb{P}(Y, S, T)$. We show that a Blackwell-optimal solution must always have an "upper-left" property—that is, the tiles associated with a positive state are located in the upper left region of this two-dimensional space. This upper-left characterization is a generalization of the result in [14], which does not require this condition. Our characterization further constrains the ratio of the widths of cells that are stacked on top of each other in the two-dimensional space based on the inferential privacy condition. This structural result enables us to substantially reduce the solution space.

(2) **Closed-form solution for binary secrets:** We obtain a closed-form expression for an inferentially-private, Blackwell-optimal information structure when the secret is binary, i.e., $|\mathcal{S}| = 2$. Notably, by virtue of Blackwell optimality, this information structure universally optimizes any decision-theoretic problem with a utility function convex in the posterior $\mathbb{P}(Y|T)$. Our analysis first makes the observation that an informative information structure will choose to maximize a subset of conditional probabilities $\mathbb{P}(T|S)$. Then to derive the optimal information structure, we analyze the dominant point of these conditional probabilities (in the sense that their values are maximized simultaneously, subject to the inferential privacy constraint, which constrains each conditional probability individually). We demonstrate that under inferential privacy constraints and Blackwell optimal structure, this dominant point exists, which enables us to derive the closed-form optimal solution and to show its uniqueness up to equivalent transformations.[1]

(3) **Lower bound on utility gains under inferential privacy (binary secrets):** When secrets are binary, we show that under a nonzero inferential privacy constraint, there exists a convex utility function such that the expected optimal utility can be arbitrarily larger than that under perfect privacy (Prop. 5.1). We demonstrate utility gains for common utility functions (e.g., quadratic), showing that by relaxing privacy constraints from perfect privacy to $\varepsilon \approx 1$, we can increase utility by up to 2×.

(4) **Program to compute optimal solution for non-binary secrets**: When there are more than two secrets (i.e., $|\mathcal{S}| > 2$), we

---

[1]We use the term *equivalent transformation* to refer to transformations that split or merge equivalent signals in terms of the posteriors $\mathbb{P}(Y|T)$ and $\mathbb{P}(S|T)$. A formal definition is provided in Definition 4.2.

provide a programming approach to compute the inferentially private information structure that is optimal for any specified utility function in the downstream decision-making problem.

*Related works.* Our work uses the definition of Inferential privacy that was formulated in Ghosh and Kleinberg [12], which measures how much information about the secret an adversary can infer from a disclosed correlated signal (precise definition in §2.1). This privacy notion is also studied in many previous works under different names, e.g, Bhaskar et al. [3], Dalenius [8], Dwork et al. [10], Kasiviswanathan and Smith [17].

Optimal information disclosure without privacy constraints has been explored extensively in the literature of Bayesian persuasion [2, 9, 15, 16]. More recently, a line of research has started the investigation of optimal information disclosure under privacy constraints. Under the perfect privacy constraint, He et al. [14] studied the informativeness of the private private signal and designed an maximally informative structure for information disclosure. Strack and Yang [23] extended He et al. [14] by considering multiple states and agents, and adopting the same form of privacy notion that requires strict independence between the output signal and secrets (albeit conditioned on auxiliary information). Their extension is complementary to ours; combining the two generalizations may be an interesting direction for future work. The perfect privacy constraint is also adopted in the worst-case information structure for auctions [1, 7]. Under differential privacy constraints [10], several works studied information disclosure [6, 13, 21]. They consider the scenarios where a third party aims to disseminate information about data collected from a number of data contributors, while protecting individual privacy, which is different to our settings.

Our work shares a similar motivation to prior work on *pufferfish privacy* [18, 22], which aims to study the privacy of correlated data, while they assume the prior distribution $\mathbb{P}(S, Y)$ is unknown to the adversary. Specifically, when the prior $\mathbb{P}(S, Y)$ is given, we show that pufferfish privacy is equivalent to inferential privacy in §2.

*Relation to He et al. [14].* Generalizing the results of He et al. [14] to accommodate an inferential privacy guarantee is not straightforward. With the generalized privacy constraints, the space of all possible information structures becomes much larger, and finding the optimal correlation between the state, the sensitive information, and the revealed signal becomes much more challenging. Our characterization only provides necessary conditions for the optimal structure, and can be viewed as an extension of the upward-closed set representation in [14] in the discrete setting. The relaxation of perfect privacy prevents us from using the classical result of Lorentz [19] about "sets of uniqueness", as in [14]. Hence, finding sufficient conditions for optimality is more challenging and Lemma 4.3 is based on entirely new techniques (details in §4.3).

## 2 PROBLEM FORMULATION

We want to maximize information disclosure about a random variable of interest $Y$ while minimizing information disclosure about a sensitive random variable $S$. The random variable of interest and the sensitive random variable are jointly drawn from a prior distribution $\mathbb{P}(S, Y)$, which is commonly known. In this work, we focus on a binary $Y \in \{0, 1\}$.

We aim to disclose an output signal that reveals information about $Y$. The output signal $T$ can be represented as a random variable that is correlated with $Y$. Our goal is to design an *information structure*, which we define as the joint probability distribution $\mathbb{P}(S, Y, T)$. This information structure has a one-to-one correspondence with our information disclosure *mechanism*, which we define as $\mathbb{P}(T|Y, S)$. We next present our privacy and utility metrics.

## 2.1 Privacy Metric

An important principle in the study of information disclosure is that "access to a statistical database should not enable one to learn anything about an individual that could not be learned without access" [8]. This qualitative notion is formalized in the definition of *inferential privacy (IP)* [12] as follows.

**Definition 2.1 (Inferential Privacy (IP)).** *An information structure $\mathbb{P}(S, Y, T)$ is $\varepsilon$-inferentially-private about $S$ if*

$$\frac{\mathbb{P}(S = s_1|T = t)}{\mathbb{P}(S = s_2|T = t)} \leq e^\varepsilon \cdot \frac{\mathbb{P}(S = s_1)}{\mathbb{P}(S = s_2)}, \quad \forall s_1, s_2 \in \mathcal{S}, t \in \mathcal{T}. \quad (1)$$

This definition can equivalently be written as

$$\frac{\mathbb{P}(T = t|S = s_1)}{\mathbb{P}(T = t|S = s_2)} \leq e^\varepsilon, \quad \forall s_1, s_2 \in \mathcal{S}, t \in \mathcal{T}. \quad (2)$$

*Relation to pufferfish privacy.* The notion of *pufferfish privacy* [18] was proposed to measure information disclosure about a secret random variable. It is defined as for all $s_1, s_2 \in \mathcal{S}, t \in \mathcal{T}, \theta \in \Theta$:

$$\mathbb{P}(T = t|S = s_1, \theta) \leq e^\varepsilon \mathbb{P}(T = t|S = s_2, \theta) + \delta, \quad (3)$$

where $\Theta$ represents the set of possible distributions $\mathbb{P}(S, Y)$. Built upon pufferfish privacy, several privacy notions, such as attribute privacy [24], are proposed by specifying $\mathcal{S}$ and $\Theta$. As we assume $\mathbb{P}(S, Y)$ is fixed and known, $\Theta$ is a singleton set. Under this condition, we note that pufferfish privacy and attribute privacy are equivalent to inferential privacy when $\delta = 0$. However, in general, pufferfish privacy can accommodate a set of distributions $\Theta$ that is not a singleton; in this case, pufferfish privacy is a stronger privacy notion than inferential privacy.

## 2.2 Informativeness and Utility

Drawing from the formulation of [14], we measure the informativeness of $T$ about $Y$ using the notion of *Blackwell ordering* [4].

**Definition 2.2 (Blackwell ordering).** *A random variable $T_1$ is more informative than $T_2$ about $Y$ if $Y \to T_1 \to T_2$ forms a Markov chain. In this case, we say information structure $\mathbb{P}(Y, T_1)$ Blackwell dominates information structure $\mathbb{P}(Y, T_2)$, and denote this condition as $\mathbb{P}(Y, T_1) \succeq \mathbb{P}(Y, T_2)$.*

Blackwell ordering has several useful properties, which are outlined in the following result.

**Theorem 2.1 (Properties of Blackwell ordering [4, 5]).** *When $Y \in \{0, 1\}$ is binary, let random variable $Q_1 = \mathbb{P}(Y = 1|T_1) \in [0, 1]$ be the posterior about $Y = 1$ after observing $T_1$. Similarly, we define $Q_2 = \mathbb{P}(Y = 1|T_2)$. Then the following statements are equivalent:*

(1) *Information structure $\mathbb{P}(Y, T_1)$ Blackwell dominates information structure $\mathbb{P}(Y, T_2)$.*

(2) *$Q_1$ is a mean-preserving spread of $Q_2$, i.e., the distribution of $Q_1$ can be derived by first taking a draw from the distribution of $Q_2$ and then adding mean-0 noise, which can depend on the draw.*

(3) *For any convex function $u$, $\mathbb{E}[u(Q_1)] \geq \mathbb{E}[u(Q_2)]$.*

From the definition of Blackwell ordering, we define $\varepsilon$-inferentially-private Blackwell optimality as follows.

**Definition 2.3 ($\varepsilon$-Inferentially-Private Blackwell optimality).** *Given $\mathbb{P}(Y, S)$, an information structure $\mathbb{P}(Y, S, T)$ is an $\varepsilon$-inferentially-private Blackwell optimal information structure if there exists no other $\varepsilon$-inferentially-private $\mathbb{P}(Y, S, T')$ that has $\mathbb{P}(Y, T') \succeq \mathbb{P}(Y, T)$.*

*Additionally, $\mathbb{P}(Y, T')$ and $\mathbb{P}(Y, T)$ are equivalent if $\mathbb{P}(Y, T') \succeq \mathbb{P}(Y, T)$ and $\mathbb{P}(Y, T) \succeq \mathbb{P}(Y, T')$.*

He et al. [14] show that for 0-inferential privacy with a binary secret, there exists a unique (up to equivalent transformations) Blackwell-optimal information structure, and they provide a closed-form expression for it (details in §3).

*Utility.* We consider a decision-theoretic formulation in which the decision maker receives a reward $r(y, a)$ under the state $y$ and their corresponding action $a$. The binary state $Y$ can be inferred from the output signal $T$ by the posterior $\mathbb{P}_{Y|T}$. We let $q_t = \mathbb{P}(Y = 1|T = t)$. Under a certain output signal $t$, we denote the utility as the maximal expected reward the decision maker can get, i.e.,

$$u(q_t) = \max_a \mathbb{E}_y[r(y, a)] = \max_a \{q_t \cdot r(1, a) + (1 - q_t) r(0, a)\}. \quad (4)$$

Since under a fixed action $a$, rewards $r(1, a)$ and $r(0, a)$ are fixed, $u(q_t)$ is a convex piecewise linear function over $q_t$. The goal of signal release mechanism design is to maximize the expected utility, i.e., $\mathbb{E}_t[u(q_t)]$, for the decision maker with any reward function.

Since Eq. (4) is convex, the result of Thm. 2.1 implies that identifying a Blackwell-optimal information structure also helps in finding a utility-maximizing structure. Indeed, we will show in §5 that when the secret is binary, the inferentially-private Blackwell-optimal structure universally maximizes the expected utility under any convex function.

## 3 GEOMETRIC VISUALIZATION OF INFORMATION STRUCTURES

As in prior work [14], we will use a visual representation of information structures to clarify the meaning of our geometric characterization. An information structure $\mathbb{P}(S, Y, T)$ can be drawn as a grid, where each column corresponds to a value of the output signal $T$, and each row corresponds to a value of the secret $S$. The color of each point in this plot denotes the posterior probability $\mathbb{P}(Y = 1|S, T)$, with white denoting $\mathbb{P}(Y = 1|S, T) = 0$ and dark yellow denoting $\mathbb{P}(Y = 1|S, T) = 1$ (light yellow denotes "in between" real values in $(0, 1)$). For example, Fig. 1 below illustrates one possible information structure with $\mathcal{S} = \{s_0, s_1\}$ and $\mathcal{T} = \{t_1, t_2, t_3\}$.

Fig. 1 fully defines the information structure: $\mathbb{P}(Y = y|S = s, T = t)$ is determined by the color of each cell, $\mathbb{P}(S = s)$ is determined by the height of row $s$, and $\mathbb{P}(T = t|S = s)$ is determined by the width of the $(s, t)$ cell. These quantities jointly determine the full distribution $\mathbb{P}(Y, S, T)$, since $\mathbb{P}(S, Y, T) = \mathbb{P}(Y|S, T) \cdot \mathbb{P}(T|S) \cdot \mathbb{P}(S)$. From the information structure, we can in turn determine the disclosure

mechanism represented by $\mathbb{P}(T|S,Y)$ because $\mathbb{P}(T|S,Y) = \frac{\mathbb{P}(S,Y,T)}{\mathbb{P}(S,Y)}$ and $\mathbb{P}(S,Y)$ is known. Since $\mathbb{P}(S)$ is known, it suffices to use $\mathbb{P}(T|S)$ and $\mathbb{P}(Y|S,T)$ to characterize a policy.

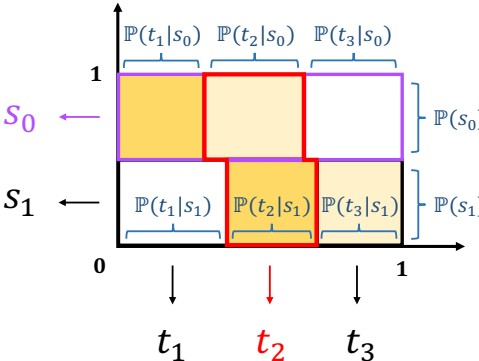

**Figure 1: Information structure of** $\mathbb{P}(S,Y,T)$. **We use the term "column" to denote a set of cells with fixed output signal** $t \in \mathcal{T}$**; in our terminology, each column need not be a single rectangle, as shown in the column outlined in red for** $t_2$**. Each row corresponds to a secret** $s \in \mathcal{S}$**. For each cell, the color represents the posterior probability** $\mathbb{P}(Y=1|S,T)$ **(dark yellow is 1, light yellow is some value between 0 and 1, and white is 0). The height of each row represents** $\mathbb{P}(S)$**, and the width of each cell represents** $\mathbb{P}(T|S)$**.**

*Interpretation of privacy constraints.* Notice that to satisfy an $\varepsilon$-inferential privacy guarantee, we need that $\frac{\mathbb{P}(T=t|S=s_1)}{\mathbb{P}(T=t|S=s_2)} \leq e^\varepsilon$ for all $t \in \mathcal{T}$. This means that in any column of the figure, the ratio of cell widths must lie in the range $[e^{-\varepsilon}, e^\varepsilon]$. Consequently, for a 0-inferential privacy constraint (as in He et al. [14]), we require that in a column, all cells have the *same* width; this implies that each column is a rectangle.

*Blackwell-optimal structure from [14].* For a binary secret and binary state $Y$, [14] provides the Blackwell-optimal information structure under a perfect privacy constraint, i.e., 0-inferential privacy, as shown in Thm. 3.1. We introduce the notation $q_s$ to denote the probability that the signal $Y=1$ given the secret signal $S=s$:

$$q_s \triangleq \mathbb{P}(Y=1|S=s).$$

Note that $q_s$ is given by the prior, and can be viewed as a constant.

**Theorem 3.1 ([14]).** *With binary state* $Y \in \{0,1\}$ *and binary secret* $S \in \{s_0, s_1\}$, *where* $\mathbb{P}(Y=1|S=s_0) \geq \mathbb{P}(Y=1|S=s_1)$, *given the joint distribution* $\mathbb{P}(S,Y)$, *the* 0-*inferentially-private Blackwell optimal information structure is unique up to equivalent transformations:* $\mathcal{T} = \{t_1, t_2, t_3\}$,

$$\mathbb{P}(T=t_1|S=s_0) = \mathbb{P}(T=t_1|S=s_1) = q_{s_1},$$
$$\mathbb{P}(T=t_2|S=s_0) = \mathbb{P}(T=t_2|S=s_1) = q_{s_0} - q_{s_1},$$
$$\mathbb{P}(T=t_3|S=s_0) = \mathbb{P}(T=t_3|S=s_1) = 1 - q_{s_0},$$
$$\mathbb{P}(Y=1|T=t_1) = \mathbb{P}(Y=1|S=s_0,T=t_2) = 1,$$
$$\mathbb{P}(Y=1|T=t_3) = \mathbb{P}(Y=1|S=s_1,T=t_2) = 0.$$

The optimal structure is visualized in Fig. 2. Under the optimal information structure, $\mathbb{P}(Y|S,T)$ is either 0 or 1, i.e., each cell is either dark yellow or white, and the dark yellow cells are in the upper left corner of the grid (we formally define 'upper left' in Definition 4.1). In this information structure, observe that the probability $\mathbb{P}(T=t|S=s)$ is the same for all values of $s$. In other words, each "column" is a rectangle in the perfect privacy setting, whose width is equal to $\mathbb{P}(T=t|S=s)$. Also note that in this example, signals $t_1$ and $t_3$ deterministically reveal $Y$: if $T=t_1$, then $Y=1$ with probability 1, whereas if $T=t_3$, then $Y=0$ with probability 1.

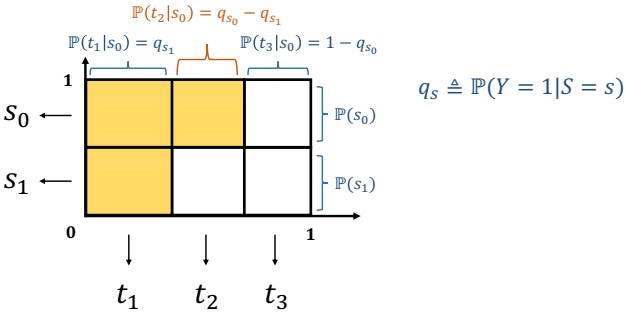

**Figure 2: Blackwell-optimal structure with perfect privacy constraint.** $\mathcal{T} = \{t_1, t_2, t_3\}$. **The width of each cell is determined by the** $\mathbb{P}(Y=1|S)$**, and each cell is either dark yellow or white, indicating** $\mathbb{P}(Y|S,T) \in \{0,1\}$**.**

## 4 GEOMETRIC CHARACTERIZATION OF IP BLACKWELL-OPTIMAL SOLUTIONS

In this section, we provide a geometric characterization of Blackwell-optimal information structures that also guarantee inferential privacy (IP). Even though the space of possible information structures is uncountably infinite, our characterization can be used to significantly limit the search space. For example, we will use the characterization to conclude that we only need to consider information structures with $|\mathcal{T}| \leq 3|\mathcal{S}| + 1$ output states (Thm. 4.1).

This characterization has three components:

(1) For any Blackwell-optimal information structure, it must hold that $\mathbb{P}(Y=1|S=s,T=t) \in \{0,1\}$. In other words, as in He et al. [14], all cells in our visualization are either white or yellow.
(2) For any output signal $t \in \mathcal{T}$ such that $\mathbb{P}(Y=1|T=t) \neq \{0,1\}$, and for any $s_1, s_2 \in \mathcal{S}$, it must hold that $\frac{\mathbb{P}(T=t|S=s_1)}{\mathbb{P}(T=t|S=s_2)} \in \{1, e^\varepsilon, e^{-\varepsilon}\}$, and the values $e^\varepsilon$ and $e^{-\varepsilon}$ can be reached by some $(s_1, s_2)$ pairs. In other words, if a column is *not* all yellow or all white, then there are exactly two cell widths in the column, and their ratio is either $e^\varepsilon$ or $e^{-\varepsilon}$, so the IP constraint is met with equality.
(3) A Blackwell-optimal structure should be "upper-left" and "lower-right", which roughly means that the yellow cells are adjacent and located in the top left corner of the visualization, and the cells with larger width, i.e., $\mathbb{P}(T=t|S=s) = \max_{s'} \mathbb{P}(T=t|S=s')$ are adjacent and located in the top left or bottom right corners of the visualization. (We state this more precisely in §4.3.)

We next discuss each of these in greater detail.

### 4.1 Geometric characterization of $\mathbb{P}(Y|S,T)$

We first show that $\mathbb{P}(Y=1|S=s,T=t)$ can only be 0 or 1 for any inferentially-private Blackwell optimal information structure.

LEMMA 4.1. *For any* $\mathbb{P}(Y, S)$ *and* $\varepsilon$, *an* $\varepsilon$-*inferentially-private Blackwell optimal information structure* $\mathbb{P}(Y, S, T)$ *must satisfy that for all* $s \in \mathcal{S}, t \in \mathcal{T}$: $\mathbb{P}(Y = 1|S = s, T = t) \in \{0, 1\}$.

The proof is in App. A. As illustrated in Fig. 3, each cell corresponds to a pair of secret $s$ and output $t$, and yellow cells indicate that $\mathbb{P}(Y = 1|S = s, T = t) = 1$, while white cells indicate that $\mathbb{P}(Y = 1|S = s, T = t) = 0$. This condition also holds for the perfect privacy setting, a special case of inferential privacy (Thm. 3.1).

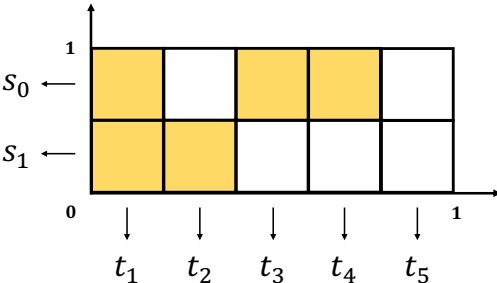

**Figure 3: Under an inferentially-private Blackwell optimal information structure,** $\mathbb{P}(Y = 1|S = s, T = t) \in \{0, 1\}, \forall s \in \mathcal{S}, t \in \mathcal{T}$. **I.e., every cell in the visualization is white or dark yellow.**

## 4.2 Geometric characterization of $\mathbb{P}(T|S)$

We first define $\widetilde{\mathcal{T}}$ as $\widetilde{\mathcal{T}} = \{t : \mathbb{P}(Y = 1|T = t) \notin \{0, 1\}\}$. For any $t \in \widetilde{\mathcal{T}}$, we analyze the relationship of $\mathbb{P}(T = t|S = s)$ for all $s$.

LEMMA 4.2. *For any* $\mathbb{P}(Y, S)$ *and* $\varepsilon$, *an* $\varepsilon$-*inferentially-private Blackwell optimal information structure* $\mathbb{P}(Y, S, T)$ *must have the inferential privacy constraints binding for any* $t \in \widetilde{\mathcal{T}}$. *Specifically, for any* $t \in \widetilde{\mathcal{T}}$, *let* $L_t = \min_s \mathbb{P}(T = t|S = s)$ *and* $H_t = \max_s \mathbb{P}(T = t|S = s)$. *We have* $H_t = e^\varepsilon \cdot L_t$, *and* $\mathbb{P}(T = t|S = s)$ *is either* $L_t$ *or* $H_t$ *for all* $s$.

The proof is in App. B. As illustrated in Fig. 4, when a column associated with output $t \in \mathcal{T}$ is neither all-yellow nor all-white, we have $\mathbb{P}(T = t|S = s) \in \{L_t, H_t\}, \forall s \in \mathcal{S}$. In other words, there are only two possible cell widths in the column, which we call "wide" and "narrow". We illustrate wide cells with red outlines to represent secret-output pairs $(s, t)$ with $\mathbb{P}(T = t|S = s) = H_t$, and use narrow cells with blue outlines to represent secret-output pairs $(s, t)$ with $\mathbb{P}(T = t|S = s) = L_t$.

## 4.3 Upper left characterization

Based on Lemmas 4.1 and 4.2, we define regions $\mathcal{A}, \mathcal{B}, \mathcal{C}$ as follows.

$$\mathcal{A} = \{(s, t) : \mathbb{P}(Y = 1|S = s, T = t) = 1, s \in \mathcal{S}, t \in \mathcal{T}\},$$

$$\mathcal{B} = \{(s, t) : (s, t) \in \mathcal{A}, \mathbb{P}(T = t|S = s) = H_t, s \in \mathcal{S}, t \in \widetilde{\mathcal{T}}\},$$

$$\mathcal{C} = \{(s, t) : (s, t) \notin \mathcal{A}, \mathbb{P}(T = t|S = s) = H_t, s \in \mathcal{S}, t \in \widetilde{\mathcal{T}}\}.$$

We next characterize upper-left and lower-right regions:

DEFINITION 4.1. *A region* $\mathcal{A}$ *is* $\mathcal{T}$-*upper-left if for any* $(s_i, t_j) \in \mathcal{A}$,

$$(s_k, t_l) \in \mathcal{A}, \quad \forall k \le i, l \le j, s_k \in \mathcal{S}, t_l \in \mathcal{T}.$$

*A region* $\mathcal{C}$ *is* $\widetilde{\mathcal{T}}$-*lower-right if for any* $(s_k, t_k) \in \mathcal{C}$,

$$(s_k, t_l) \in \mathcal{C}, \quad \forall k \ge i, l \ge j, s_k \in \mathcal{S}, t_l \in \widetilde{\mathcal{T}}.$$

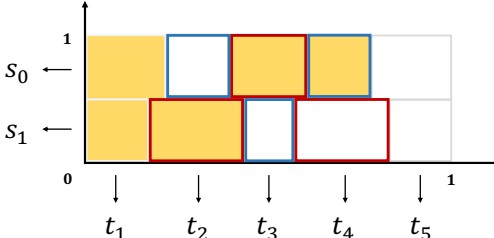

**Figure 4: Under an inferentially-private Blackwell-optimal information structure with** $\mathcal{S} = \{s_0, s_1\}$ **and** $\widetilde{\mathcal{T}} = \{t_2, t_3, t_4\}$, $\mathbb{P}(T = t_2|S = s_1) = H_{t_2}$, $\mathbb{P}(T = t_3|S = s_0) = H_{t_3}$, $\mathbb{P}(T = t_4|S = s_1) = H_{t_4}$, **illustrated by cells with red outlines,** $\mathbb{P}(T = t_2|S = s_0) = L_{t_2}$, $\mathbb{P}(T = t_3|S = s_1) = L_{t_3}$, $\mathbb{P}(T = t_4|S = s_0) = L_{t_4}$, **illustrated by cells with blue outlines, and** $H_t = e^\varepsilon \cdot L_t, \forall t \in \{t_2, t_3, t_4\}$.

Intuitively, upper-left means that starting from any cell in the region, every cell above it and/or to the left is also part of the region. Similarly, lower-right means that for any cell in the region, all cells below and/or to the right of the cell are also in the region.

LEMMA 4.3. *Consider any* $\varepsilon$-*inferentially-private Blackwell optimal information structure* $\mathbb{P}(Y, S, T)$. *Suppose* $s_1, \ldots, s_n \in \mathcal{S}$ *are ordered in decreasing order of* $\mathbb{P}(Y = 1|S = s)$, *and* $t_1, \ldots, t_k \in \mathcal{T}$ *are ordered in decreasing order of* $\mathbb{P}(Y = 1|T = t)$. *Then the region* $\mathcal{A}$ *is* $\mathcal{T}$-*upper-left, region* $\mathcal{B}$ *is* $\widetilde{\mathcal{T}}$-*upper-left, and region* $\mathcal{C}$ *is* $\widetilde{\mathcal{T}}$-*lower-right.*

The proof is in App. C. An example is shown in Fig. 5; the yellow cells form the region $\mathcal{A}$, the cells in yellow with red outlines form the region $\mathcal{B}$, and cells in white with red outlines form the region $\mathcal{C}$. The region $\mathcal{A}$ is $\mathcal{T}$-upper-left, region $\mathcal{B}$ is $\widetilde{\mathcal{T}}$-upper-left, and region $\mathcal{C}$ is $\widetilde{\mathcal{T}}$-lower-right. The upper-left characterization of region $\mathcal{A}$ is also true in the perfect privacy setting, while the characterizations of regions $\mathcal{B}$ and $\mathcal{C}$ are specific to the inferential privacy setting.

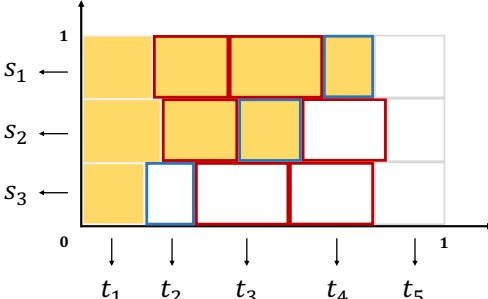

**Figure 5: Under an inferentially-private Blackwell optimal information structure with** $\mathcal{S} = \{s_1, s_2, s_3\}$ **and** $\widetilde{\mathcal{T}} = (t_2, t_3, t_4)$, **region** $\mathcal{A}$ **is illustrated as the yellow cells, region** $\mathcal{B}$ **is illustrated as the yellow cells with red outlines, and region** $\mathcal{C}$ **is illustrated as the white cells with red outlines. The region** $\mathcal{A}$ **is** $\mathcal{T}$-**upper-left, region** $\mathcal{B}$ **is** $\widetilde{\mathcal{T}}$-**upper-left, and region** $\mathcal{C}$ **is** $\widetilde{\mathcal{T}}$-**lower-right.**

*Remark on proof techniques.* Lemma 4.3 can be viewed as an extension of the upward-closed set representation in [14, Theorem 3] in the discrete setting. The relaxation of the privacy constraint

($\varepsilon > 0$) introduces a fundamental difference to the underlying geometry and the previous technique of He et al. [14] for the perfect privacy setting, which uses the classical result on sets of uniqueness by Lorentz [19], can no longer be applied. As a result, finding the sufficient condition for optimality is much more challenging and Lemma 4.3 is based on entirely new techniques. The key idea is to identify "microstructures" that cannot appear in an optimal structure. We prove that information structures without such "microstructures" must have $S$ and $T$ that can be reordered to exhibit a structure that can be represented by three upward-closed sets, as opposed to one in the perfect privacy case.

### 4.4 Cardinality of the Output Signal Set

The geometric characterization of the $\varepsilon$-inferentially-private Blackwell optimal information structure allows us to upper bound the number of outputs needed to construct an optimal information structure.[2] We first define the equivalent transformation as follows.

**Definition 4.2 (Equivalent Transformation).** *The equivalent transformation of an information structure* $\mathbb{P}(S, Y, T)$ *consists of one or multiple operations shown as follows. Denote* $\hat{\mathbb{P}}\left(Y, S, \hat{T}\right)$ *as the information structure after transformation.*

- **Split**. *Split an output signal* $t_i$ *into a set of equivalent signals* $\hat{\mathcal{T}}_i$ *that share the same geometric pattern:*

$$\mathbb{P}(T = t_i) = \sum_{\hat{t} \in \hat{\mathcal{T}}_i} \hat{\mathbb{P}}\left(\hat{T} = \hat{t}\right),$$

$$\mathbb{P}(Y = 1 | T = t_i) = \hat{\mathbb{P}}\left(Y = 1 | \hat{T} = \hat{t}\right), \quad \forall \hat{t} \in \hat{\mathcal{T}}_i,$$

$$\mathbb{P}(S = s | T = t_i) = \hat{\mathbb{P}}\left(S = s | \hat{T} = \hat{t}\right), \quad \forall s \in \mathcal{S}, \ \hat{t} \in \hat{\mathcal{T}}_i.$$

- **Merge**. *Merge a set of signals* $\mathcal{T}_i$ *to an equivalent signal* $\hat{t}_i$ *that shares the same geometric pattern:*

$$\sum_{t \in \mathcal{T}_i} \mathbb{P}(T = t) = \hat{\mathbb{P}}\left(\hat{T} = \hat{t}_i\right),$$

$$\mathbb{P}(Y = 1 | T = t) = \hat{\mathbb{P}}\left(Y = 1 | \hat{T} = \hat{t}_i\right), \quad \forall t \in \mathcal{T}_i,$$

$$\mathbb{P}(S = s | T = t) = \hat{\mathbb{P}}\left(S = s | \hat{T} = \hat{t}_i\right), \quad \forall s \in \mathcal{S}, \ t \in \mathcal{T}_i.$$

We say that $\hat{\mathbb{P}}\left(Y, S, \hat{T}\right)$ is an equivalent information structure to $\mathbb{P}(Y, S, T)$ if it can be obtained by equivalent transformation.

**Theorem 4.1.** *Given* $\mathbb{P}(S, Y)$ *and* $\varepsilon > 0$*, for any* $\varepsilon$*-inferentially-private Blackwell optimal information structure, there exists an equivalent information structure* $\mathbb{P}(Y, S, T)$ *that has* $|\mathcal{T}| \leq 3|\mathcal{S}| + 1$*.*

The proof is shown in App. D.

## 5 MECHANISM DESIGN: BINARY SECRET

The geometric characterization of inferentially-private Blackwell optimal information structures significantly reduces the search space for solutions. However, it does not allow us to trivially determine a Blackwell-optimal solution in general. We next design an

---

[2]Note that the analysis of the geometric characterization in Lemmas 4.1 to 4.3 does not rely on the existence of a finite-size Blackwell optimal structure.

information disclosure mechanism that achieves an $\varepsilon$-inferentially-private Blackwell optimal information structure when the secret is binary, i.e., $\mathcal{S} = \{s_0, s_1\}$. We provide an optimal disclosure mechanism that is closed-form, only uses 4 output signals, and is unique up to equivalent transformations. The designed mechanism universally maximizes the expected utility $\mathbb{E}_t[u(q_t)]$, where $q_t = \mathbb{P}(Y = 1 | T = t)$, of the decision maker under any reward function.

We defer the analysis of general secrets with $n > 2$ possible values, i.e., $\mathcal{S} = \{s_1, \dots, s_n\}$, to App. I. The main result is to derive a set of programs that lead to an optimal solution for any given utility function in the downstream decision-making problem. The design of the programs depends on our geometric characterization, which ensures that each program is linear.

### 5.1 Geometric characterization: binary secret

We first start by presenting the geometric characterization under the special case of a binary secret. Let $q_s = \mathbb{P}(Y = 1 | S = s)$, $q_t = \mathbb{P}(Y = 1 | T = t)$ and $p_t = \mathbb{P}(T = t)$. Denote $l_i^{(j)} = \mathbb{P}(T = t_i | S = s_j)$, where $j \in \{0, 1\}$, and $\boldsymbol{l} = \left\{l_i^{(j)}\right\}_{i \in |\mathcal{T}|, j \in \{0,1\}}$. In other words, $l_i^{(j)}$ is the width of the cell in the $i$th column and the $j$th row. As discussed in §3, we can fully determine the information structure of $\mathbb{P}(S, Y, T)$ by specifying the values of $l_i^{(j)}$ and $\mathbb{P}(Y | S, T)$, and in turn determines the disclosure mechanism. The following lemma shows the characterization of a Blackwell-optimal information structure $\mathbb{P}(Y, S, T)$ by specifying the constraints on values of $l_i^{(j)}$, which in turn can specify $\mathbb{P}(Y | S, T)$ together with Lemmas 4.1 and 4.3.

**Lemma 5.1 (Feasibility condition).** *Given* $\mathbb{P}(S, Y)$ *and* $\varepsilon > 0$*, for any* $\varepsilon$*-inferentially-private Blackwell optimal information structure with* $\mathcal{S} = \{s_0, s_1\}$*, there exists an equivalent information structure* $\mathbb{P}(Y, S, T)$ *that satisfies* $|\mathcal{T}| \leq 4$ *and its associated* $\boldsymbol{l}$ *values have the following properties:*

- $l_1^{(1)}$ *and* $l_4^{(0)}$ *are fixed to be:*

$$l_1^{(1)} = q_{s_1}, \quad l_4^{(0)} = 1 - q_{s_0}, \tag{5}$$

- $\boldsymbol{l}$ *ensure the disclosure policy satisfies the IP constraint with:*

$$\frac{l_2^{(0)}}{l_2^{(1)}} = \frac{l_3^{(1)}}{l_3^{(0)}} = e^\varepsilon, \tag{6}$$

$$l_1^{(0)} \in \left[e^{-\varepsilon} l_1^{(1)}, e^\varepsilon l_1^{(1)}\right], \quad l_4^{(1)} \in \left[e^{-\varepsilon} l_4^{(0)}, e^\varepsilon l_4^{(0)}\right], \tag{7}$$

- $\boldsymbol{l}$ *are valid probabilities:*

$$\sum_{i \in [4]} l_i^{(j)} = 1, \quad \forall j \in \{0, 1\}, \tag{8}$$

$$l_i^{(j)} \geq 0, \quad \forall i \in [4], j \in \{0, 1\}. \tag{9}$$

The proof is shown in App. E. The geometric characterization of the optimal structure with binary secret is illustrated in Fig. 6. For the cell that corresponds to the secret $s_i$ and the output $t_j$, $\forall i \in \{0, 1\}, j \in [4]$, the vertical length represents $\mathbb{P}(S = s_i)$ and the horizontal length represents $l_j^{(i)} = \mathbb{P}(T = t_j | S = s_i)$.

Recall that our geometric characterization of Blackwell-optimal solutions implies that, as in the case of perfect inferential privacy, the information structure must satisfy the property that each cell

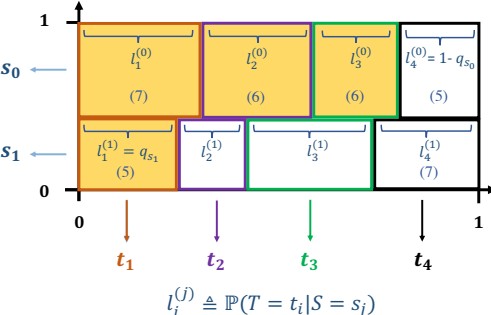

$$l_i^{(j)} \triangleq \mathbb{P}(T = t_i | S = s_j)$$

**Figure 6: Geometric characterization of an $\varepsilon$-inferentially-private Blackwell optimal information structure with a binary secret. There are at most four output signals $t_1, t_2, t_3, t_4$; signals $t_1$ and $t_4$ completely reveal the signal $Y$. For shorthand, we use $l_i^{(j)}$ to represent $\mathbb{P}\left(T = t_i | S = s_j\right)$. Cells with the same border color correspond to the same output signal $T$. The widths of the cells in the bottom-left and top-right corners are fully determined by Eq. (5) in Lemma 5.1: namely, $l_1^{(1)} = q_{s_1}$ and $l_4^{(0)} = 1 - q_{s_0}$. The ratio between the widths of the cells with purple or green outlines satisfies $\frac{l_2^{(0)}}{l_2^{(1)}} = \frac{l_3^{(1)}}{l_3^{(0)}} = e^\varepsilon$, according to Eq. (6). The widths of the cells in top-left and bottom-right corners satisfy $l_1^{(0)} \in \left[e^{-\varepsilon} l_1^{(1)}, e^\varepsilon l_1^{(1)}\right], l_4^{(1)} \in \left[e^{-\varepsilon} l_4^{(0)}, e^\varepsilon l_4^{(0)}\right]$ respectively, according to Eq. (7).**

is either dark yellow or white, i.e., $\mathbb{P}(Y|S,T) \in \{0,1\}$, and the dark yellow cells are in the upper left region of the grid. Further, Eq. (5) implies that the bottom left and top right cells have a width that is fixed and independent of the IP constraint. Note that, as with the perfect privacy result of He et al. [14], the outermost signals $t_1$ and $t_4$ fully reveal the state $Y$. However, recall that the perfect privacy case required at most three output signals. Lemma 5.1 implies that four output signals are sufficient; we show in §5.2 that in some cases, four output signals are also necessary. Moreover, while the inner columns for $t_2$ and $t_3$ have a cell width ratio that is exactly $e^\varepsilon$ (thus meeting the IP constraint with equality), the first and fourth columns have a ratio of cell widths that does not necessarily meet the IP constraint with equality. Hence, the main difficulty of finding a Blackwell-optimal mechanism is to identify the exact width and cell width ratio for the first and fourth columns.

## 5.2 Mechanism design

We split this section into two steps. First, we present our main result for the binary case, which is a Blackwell-optimal information structure under an $\varepsilon$-IP constraint. Then we demonstrate how this information structure leads to an information disclosure mechanism, and analyze its utility compared to an optimal mechanism under a perfect privacy constraint.

*5.2.1 Step 1: Determine the Blackwell-Optimal Solution.* Recall that our geometric characterization in Lemma 5.1 fixed the size of the bottom left and top right cells of any Blackwell-optimal structure (i.e., bottom orange cell and top black cell in Fig. 6). Next, we specify the sizes of the other cells. To maximize utility, we want the top

left and bottom right cells, i.e., $l_1^{(0)}$ and $l_4^{(1)}$, to be as wide as possible, since they deterministically reveal $Y$. In principle, these two cell widths could depend on each other. For instance, increasing $l_1^{(0)}$ might force $l_4^{(1)}$ to shrink, for the solution to remain feasible. However, in this subsection, we show that this is not the case: $l_1^{(0)}$ and $l_4^{(1)}$ can be maximized simultaneously.

We first show formally that in order to maximize expected utility, we want $l_1^{(0)}$ and $l_4^{(1)}$ to be as large as possible. We define the feasible dominant point as follows.

DEFINITION 5.1 (FEASIBLE DOMINANT POINT). *$l$ is a feasible dominant point if it satisfies the feasibility constraints in Lemma 5.1 and if for any point $l'$ that satisfies the feasibility constraints, it holds that $l_1^{(0)} \geq l'_1^{(0)}$ and $l_4^{(1)} \geq l'_4^{(1)}$.*

The following lemma shows that there exists a unique feasible dominant point, and it maximizes expected utility under any convex utility function.

LEMMA 5.2. *There exists a unique feasible dominant point. For any convex utility function $u$, the objective function $\mathbb{E}_t[u(q_t)]$ is maximized if and only if $l$ is the feasible dominant point.*

The proof is shown in App. F. Combining with Thm. 2.1, we know that the inferentially-private Blackwell optimal information structure is determined by the feasible dominant point. We next provide the unique inferentially-private Blackwell optimal information structure. This structure is unique up to transformations of the structure that merge equivalent output signals.

THEOREM 5.1. *Let $R_1 = \frac{q_{s_0}}{q_{s_1}}$ and $R_2 = \frac{1 - q_{s_1}}{1 - q_{s_0}}$. Given the joint distribution $\mathbb{P}(S, Y)$ and $\varepsilon$, the following information structure is the unique $\varepsilon$-inferentially-private Blackwell-optimal information structure (unique up to equivalent transformations) that universally maximizes the expected utility under any convex utility function: $l$ satisfies conditions in Lemma 5.1 , and*

| *When:* | *Then:* |
|---|---|
| $R_1 \leq e^\varepsilon, R_2 \leq e^\varepsilon$ | $l_2^{(1)} = l_3^{(1)} = 0$ |
| $R_1 \leq e^\varepsilon, R_2 > e^\varepsilon$, or $R_1 > e^\varepsilon, R_2 > e^\varepsilon, q_{s_1} \geq \frac{1}{1+e^\varepsilon}$ | $l_2^{(1)} = 0,$ $l_3^{(1)} = 1 - q_{s_1} - e^\varepsilon(1 - q_{s_0})$ |
| $R_1 > e^\varepsilon, R_2 \leq e^\varepsilon$, or $R_1 > e^\varepsilon, R_2 > e^\varepsilon, q_{s_0} \leq \frac{1}{1+e^{-\varepsilon}}$ | $l_2^{(1)} = e^{-\varepsilon} q_{s_0} - q_{s_1},$ $l_3^{(1)} = 0$ |
| $R_1 > e^\varepsilon, R_2 > e^\varepsilon, q_{s_0} > \frac{1}{1+e^{-\varepsilon}}, q_{s_1} < \frac{1}{1+e^\varepsilon}$ | $l_2^{(1)} = \frac{1}{e^\varepsilon+1} - q_{s_1},$ $l_3^{(1)} = e^\varepsilon q_{s_0} - \frac{e^{2\varepsilon}}{e^\varepsilon+1}$ |

The proof is shown in App. G. As illustrated in Fig. 7, there are six regions of problem parameters, which depend on both the prior and $\varepsilon$, that determine how many (and which) signals we need. When $R_1 \leq e^\varepsilon, R_2 \leq e^\varepsilon$, i.e., the original secret-state structure $\mathbb{P}(S, Y)$ already satisfies the inferential privacy constraint, we can just release the actual state and therefore $T = \{t_1, t_4\}$. When $R_1 \leq e^\varepsilon, R_2 > e^\varepsilon$ or $R_1 > e^\varepsilon, R_2 \leq e^\varepsilon$, the original secret-state structure $\mathbb{P}(S, Y)$ satisfies the inferential privacy constraint only when $Y = 1$ or $Y = 0$, and we need to introduce an additional signal to ensure the inferential privacy constraint is met. When $R_1 > e^\varepsilon, R_2 > e^\varepsilon$, i.e., the original secret-state structure $\mathbb{P}(S, Y)$ does not satisfy the inferential privacy constraint, the number of output signal required

depends on the value of $q_{s_1}$ or $q_{s_0}$. When $q_{s_1} \geq \frac{1}{1+e^\varepsilon}$ or $q_{s_0} \leq \frac{1}{1+e^{-\varepsilon}}$, only three signals are required. Otherwise, we need four output signals to achieve a Blackwell optimal structure under the inferential privacy constraint.

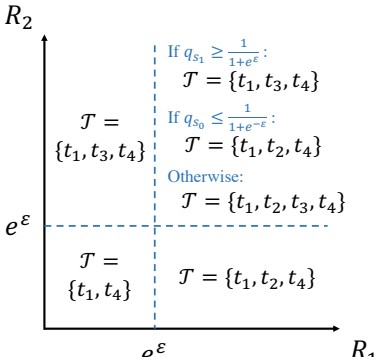

**Figure 7: Output signal set $\mathcal{T}$ with different $R_1, R_2$. When $R_1 \leq e^\varepsilon, R_2 \leq e^\varepsilon$, $\mathbb{P}(S,Y)$ already satisfies the inferential privacy constraint, and we can just release the actual state with two output signals. When $R_1 \leq e^\varepsilon, R_2 > e^\varepsilon$ or $R_1 > e^\varepsilon, R_2 \leq e^\varepsilon$, $\mathbb{P}(S,Y)$ satisfies the inferential privacy constraint only when $Y = 1$ or $Y = 0$, and an additional signal is required. When $R_1 > e^\varepsilon, R_2 > e^\varepsilon$, the number of output signal required depends on the value of $q_{s_1}$ or $q_{s_0}$.**

#### 5.2.2 Step 2: Determine the information disclosure mechanism.
Based on the inferentially-private Blackwell optimal information structure in Thm. 5.1, and the fact that $\mathbb{P}(S,Y,T) = \mathbb{P}(Y|S,T) \cdot \mathbb{P}(T|S) \cdot \mathbb{P}(S)$ and $\mathbb{P}(T|S,Y) = \frac{\mathbb{P}(S,Y,T)}{\mathbb{P}(S,Y)}$, we can design the universally optimal mechanism, represented by $\mathbb{P}(T|S,Y)$, as in Corollary 5.1.

**Corollary 5.1.** *Given $\mathbb{P}(S,Y)$ and $\varepsilon$, the universally optimal $\varepsilon$-inferentially-private mechanism that maximizes the objective function $\mathbb{E}_t[u(q_t)]$ under any convex utility function $u$ is unique up to equivalent transformations:*

$$\mathbb{P}(T = t_1 | S = s_1, Y = 1) = 1,$$

$$\mathbb{P}(T = t_i | S = s_0, Y = 1) = l_i^{(0)} / q_{s_0}, \quad \forall i \in \{1, 2, 3\},$$

$$\mathbb{P}(T = t_4 | S = s_0, Y = 0) = 1,$$

$$\mathbb{P}(T = t_i | S = s_1, Y = 0) = l_i^{(1)} / (1 - q_{s_1}), \quad \forall i \in \{2, 3, 4\},$$

*where the values of $l$ are shown in Thm. 5.1.*

The universally optimal mechanism is unique given $\mathbb{P}(S,Y)$ and $\varepsilon$, and fully reveals the state when $S = s_1, Y = 1$ or $S = s_0, Y = 0$.

#### 5.2.3 Utility gains under inferential privacy.
In general, relaxing perfect privacy to $\varepsilon$-IP can lead to significant utility gains. We can show that under an inferential privacy constraint of $\varepsilon > 0$, there exists a Lipschitz-continuous, convex utility function such that the maximal expected utility is arbitrarily larger than it would have been under perfect privacy constraints, i.e., with IP level $\varepsilon = 0$.

**Proposition 5.1.** *Denote the maximal achievable expected utility under perfect privacy constraint as $U_0$, and the maximal achievable expected utility under inferential privacy constraint $\varepsilon > 0$ as $U_\varepsilon$. For any $\varepsilon > 0, \Delta \in \mathbb{R}$, there exists a joint distribution $\mathbb{P}(S,Y)$ and a*

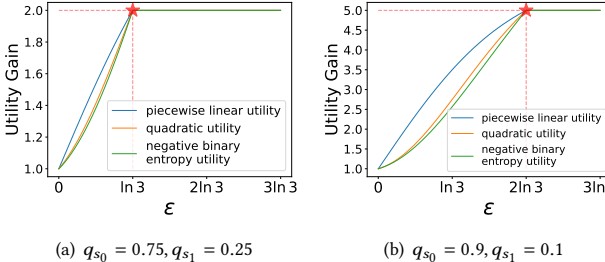

(a) $q_{s_0} = 0.75, q_{s_1} = 0.25$      (b) $q_{s_0} = 0.9, q_{s_1} = 0.1$

**Figure 8: Expected utility gain under the optimal mechanism for three utility functions and inferential privacy constraints $\varepsilon$. By relaxing $\varepsilon$ to $\ln 3$ or $2 \ln 3$, the utility of the optimal mechanism at $\varepsilon = 0$ can be improved by up to $5\times$, depending on the data distribution.**

*$L$-Lipschitz convex utility function $u$, where $L \leq 3\Delta \left(1 + \frac{2}{e^\varepsilon - 1}\right)$, such that $U_\varepsilon - U_0 \geq \Delta$.*

The proof is shown in App. H. We illustrate the utility gain of relaxing a perfect privacy constraint to an $\varepsilon$-IP constraint for some examples in Fig. 8. We define *utility gain* as the ratio of the maximal expected utility under inferential privacy constraint $\varepsilon$ to the utility under a perfect privacy constraint, i.e., $\varepsilon = 0$. We set $\mathbb{P}(S = s_0) = \mathbb{P}(S = s_1) = 0.5$ and vary $q_{s_0}, q_{s_1}$ in Figs. 8(a) and 8(b). We consider three convex utility functions $u(q_t)$ including piecewise linear $u(q_t) = |2q_t - 1|$, quadratic $u(q_t) = (2q_t - 1)^2$, and the (shifted) negative binary entropy function $u(q_t) = q_t \log q_t + (1 - q_t) \log(1 - q_t) + 1$.

Figure 8 shows that when $q_s$ is imbalanced, relaxing inferential privacy to a level of $\varepsilon = \ln 3$ (left) or $\varepsilon = 2 \ln 3$ (right) can give utility gains of $2\times$ and $5\times$, respectively. In other words, even under relatively strong privacy parameters, a small relaxation in privacy can give a significant gain in utility.

## 6 CONCLUSION

In this work, we generalize the private private information structures of He et al. [14] from a perfect privacy constraint to an inferentially-private privacy constraint. To devise a Blackwell optimal disclosure mechanism under such an inferential privacy constraint, we first derive a geometric characterization of the corresponding optimal information structure. This characterization facilitates exact analysis in special cases. In the binary secret setting, we obtain a closed-form expression for an inferentially-private Blackwell-optimal information structure, which is universally optimal in the sense that it maximizes the expected utility under any convex utility function. We finally provide a programming approach to compute the optimal solution for a specified utility function when the secret is nonbinary.

Our work leaves several important questions unanswered. For example, in the case of general (non-binary) secrets, it is unclear how to derive a closed-form expression for a Blackwell-optimal mechanism. Another important assumption we have made is that the prior $\mathbb{P}(Y,S)$ is known *a priori*. Understanding how to relax this assumption (thereby approaching a privacy definition akin to pufferfish privacy) while still providing optimality guarantees is an interesting direction.

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

# APPENDIX

## A   PROOF OF LEMMA 4.1

Proof. For a distribution $\mathbb{P}(Y, S, T)$ that has $\mathbb{P}(Y = 1 | S = s, T = t) = k \in (0, 1)$ for some $s$ and $t$, we can split $t$ into two signals $t_1$ and

$t_2$ such that

$$\mathbb{P}_{\text{new}}(Y = 1 | S = s, T = t_1) = 1,$$
$$\mathbb{P}_{\text{new}}(Y = 1 | S = s, T = t_2) = 0,$$
$$\mathbb{P}_{\text{new}}(Y = 1 | S = s', T = t_1) = \mathbb{P}_{\text{new}}(Y = 1 | S = s', T = t_2)$$
$$= \mathbb{P}(Y = 1 | S = s', T = t), \quad \forall s' \neq s,$$

while keeping the posterior of $S$ after seeing $t_1$ or $t_2$ the same as $\mathbb{P}(S | T = t)$ by letting

$$\mathbb{P}_{\text{new}}(S = s', T = t_1) = \mathbb{P}(S = s', T = t) \cdot \frac{\mathbb{P}_{\text{new}}(S = s, T = t_1)}{\mathbb{P}(S = s, T = t)},$$
$$\mathbb{P}_{\text{new}}(S = s', T = t_2) = \mathbb{P}(S = s', T = t) \cdot \frac{\mathbb{P}_{\text{new}}(S = s, T = t_2)}{\mathbb{P}(S = s, T = t)}.$$

We can get that

$$\mathbb{P}_{\text{new}}(Y = 1 | T = t_1) = \mathbb{P}_{\text{new}}(Y = 1 | S = s, T = t_1) \cdot \mathbb{P}_{\text{new}}(S = s | T = t_1)$$
$$+ \sum_{s' \neq s} \mathbb{P}_{\text{new}}(Y = 1 | S = s', T = t_1) \cdot \mathbb{P}_{\text{new}}(S = s' | T = t_1)$$
$$= \mathbb{P}(S = s | T = t) + \sum_{s' \neq s} \mathbb{P}(Y = 1 | S = s', T = t) \cdot \mathbb{P}(S = s' | T = t)$$
$$= \mathbb{P}(Y = 1 | T = t) + (1 - k) \mathbb{P}(S = s | T = t)$$
$$> \mathbb{P}(Y = 1 | T = t),$$

as well as

$$\mathbb{P}_{\text{new}}(Y = 1 | T = t_2) = \mathbb{P}_{\text{new}}(Y = 1 | S = s, T = t_2) \cdot \mathbb{P}_{\text{new}}(S = s | T = t_2)$$
$$+ \sum_{s' \neq s} \mathbb{P}_{\text{new}}(Y = 1 | S = s', T = t_2) \cdot \mathbb{P}_{\text{new}}(S = s' | T = t_2)$$
$$= \sum_{s' \neq s} \mathbb{P}(Y = 1 | S = s', T = t) \cdot \mathbb{P}(S = s' | T = t)$$
$$= \mathbb{P}(Y = 1 | T = t) - k \cdot \mathbb{P}(S = s | T = t)$$
$$< \mathbb{P}(Y = 1 | T = t).$$

Therefore, $\mathbb{P}_{\text{new}}(Y = 1 | T)$ is a mean-preserving spread of $\mathbb{P}(Y = 1 | T)$. From Thm. 2.1, we know that the new $\mathbb{P}_{\text{new}}(Y, T_{\text{new}})$ Blackwell dominates $\mathbb{P}(Y, T)$, but the reverse is not true; and the privacy guarantee is preserved as the posterior of $S$ after seeing $t_1$ or $t_2$ is the same as $\mathbb{P}(S | T = t)$. Therefore a distribution with $\mathbb{P}(Y = 1 | S = s, T = t) \in (0, 1)$ cannot be an inferentially-private Blackwell optimal information structure. □

## B   PROOF OF LEMMA 4.2

Proof. Based on inferential privacy constraint, we know that $\forall t, s_1, s_2, \frac{\mathbb{P}(T=t|S=s_1)}{\mathbb{P}(T=t|S=s_2)} \in [e^{-\varepsilon}, e^{\varepsilon}]$. Therefore, we can get that

$$\frac{H_t}{L_t} = \frac{\max_s \mathbb{P}(T = t | S = s)}{\min_s \mathbb{P}(T = t | S = s)} \in [1, e^{\varepsilon}].$$

Suppose there exists $t$ with $\mathbb{P}(Y = 1 | T = t) \in (0, 1)$ and $\mathbb{P}(T = t | S = s) \in (L_t, e^{\varepsilon} L_t)$ for some $s$, we can split $t$ into two signals $t_1$ and $t_2$ such that

$$\mathbb{P}_{\text{new}}(T = t_1 | S = s) = \frac{1}{2}\mathbb{P}(T = t | S = s) + \delta,$$
$$\mathbb{P}_{\text{new}}(T = t_2 | S = s) = \frac{1}{2}\mathbb{P}(T = t | S = s) - \delta,$$
$$\forall s' \neq s:$$
$$\mathbb{P}_{\text{new}}(T = t_1 | S = s') = \mathbb{P}_{\text{new}}(T = t_2 | S = s') = \frac{1}{2}\mathbb{P}(T = t | S = s'),$$

where $\delta \in \left(0, \min\left\{\frac{1}{2}\mathbb{P}(T = t|S = s) - \frac{1}{2}L_t, \frac{e^\epsilon}{2}L_t - \frac{1}{2}\mathbb{P}(T = t|S = s)\right\}\right]$.
We can easily check that the constructed structure $\mathbb{P}_{new}(Y, S, T)$ satisfies the inferential privacy constraints. Based on Lemma 4.1, we know that $\forall s_0 \in \mathcal{S} : \mathbb{P}(Y = 1|T = t, S = s_0) = \mathbb{P}_{new}(Y = 1|T = t_1, S = s_0) = \mathbb{P}_{new}(Y = 1|T = t_2, S = s_0) \in \{0, 1\}$. Since $\mathbb{P}(Y = 1|T = t) \in (0, 1)$, when $\mathbb{P}(Y = 1|T = t, S = s) = 1$, we can get that

$$\mathbb{P}_{new}(Y = 1|T = t_1)$$

$$= \mathbb{P}_{new}(Y = 1|S = s, T = t_1) \cdot \frac{\mathbb{P}_{new}(T = t_1|S = s) \cdot \mathbb{P}(S = s)}{\sum_{s_0 \in \mathcal{S}} \mathbb{P}_{new}(T = t_1|S = s_0) \cdot \mathbb{P}(S = s_0)}$$

$$+ \sum_{s' \neq s} \mathbb{P}_{new}(Y = 1|S = s', T = t_1) \cdot \frac{\mathbb{P}_{new}(T = t_1|S = s') \cdot \mathbb{P}(S = s')}{\sum_{s_0 \in \mathcal{S}} \mathbb{P}_{new}(T = t_1|S = s_0) \cdot \mathbb{P}(S = s_0)}$$

$$= \mathbb{P}(Y = 1|S = s, T = t) \cdot \frac{\frac{1}{2}\mathbb{P}(T = t|S = s) \cdot \mathbb{P}(S = s) + \delta\mathbb{P}(S = s)}{\frac{1}{2}\mathbb{P}(T = t) + \delta\mathbb{P}(S = s)}$$

$$+ \sum_{s' \neq s} \mathbb{P}(Y = 1|S = s', T = t) \cdot \frac{\frac{1}{2}\mathbb{P}(T = t|S = s') \cdot \mathbb{P}(S = s')}{\frac{1}{2}\mathbb{P}(T = t) + \delta\mathbb{P}(S = s)}$$

$$= \mathbb{P}(Y = 1|T = t) \cdot \frac{\mathbb{P}(T = t)}{\mathbb{P}(T = t) + 2\delta \cdot \mathbb{P}(S = s)} + \frac{2\delta \cdot \mathbb{P}(S = s)}{\mathbb{P}(T = t) + 2\delta \cdot \mathbb{P}(S = s)}$$

$$> \mathbb{P}(Y = 1|T = t),$$

as well as

$$\mathbb{P}_{new}(Y = 1|T = t_2)$$

$$= \mathbb{P}_{new}(Y = 1|S = s, T = t_2) \cdot \frac{\mathbb{P}_{new}(T = t_2|S = s) \cdot \mathbb{P}(S = s)}{\sum_{s_0 \in \mathcal{S}} \mathbb{P}_{new}(T = t_2|S = s_0) \cdot \mathbb{P}(S = s_0)}$$

$$+ \sum_{s' \neq s} \mathbb{P}_{new}(Y = 1|S = s', T = t_2) \cdot \frac{\mathbb{P}_{new}(T = t_2|S = s') \cdot \mathbb{P}(S = s')}{\sum_{s_0 \in \mathcal{S}} \mathbb{P}_{new}(T = t_2|S = s_0) \cdot \mathbb{P}(S = s_0)}$$

$$= \mathbb{P}(Y = 1|S = s, T = t) \cdot \frac{\frac{1}{2}\mathbb{P}(T = t|S = s) \cdot \mathbb{P}(S = s) - \delta\mathbb{P}(S = s)}{\frac{1}{2}\mathbb{P}(T = t) - \delta\mathbb{P}(S = s)}$$

$$+ \sum_{s' \neq s} \mathbb{P}(Y = 1|S = s', T = t) \cdot \frac{\frac{1}{2}\mathbb{P}(T = t|S = s') \cdot \mathbb{P}(S = s')}{\frac{1}{2}\mathbb{P}(T = t) - \delta\mathbb{P}(S = s)}$$

$$= \mathbb{P}(Y = 1|T = t) \cdot \frac{\mathbb{P}(T = t)}{\mathbb{P}(T = t) - 2\delta \cdot \mathbb{P}(S = s)} - \frac{2\delta \cdot \mathbb{P}(S = s)}{\mathbb{P}(T = t) - 2\delta \cdot \mathbb{P}(S = s)}$$

$$< \mathbb{P}(Y = 1|T = t).$$

When $\mathbb{P}(Y = 1|T = t, S = s) = 1$, similarly, we can get that

$$\mathbb{P}_{new}(Y = 1|T = t_1)$$

$$= \sum_{s' \neq s} \mathbb{P}(Y = 1|S = s', T = t) \cdot \frac{\frac{1}{2}\mathbb{P}(T = t|S = s') \cdot \mathbb{P}(S = s')}{\frac{1}{2}\mathbb{P}(T = t) + \delta\mathbb{P}(S = s)}$$

$$= \mathbb{P}(Y = 1|T = t) \cdot \frac{\mathbb{P}(T = t)}{\mathbb{P}(T = t) + 2\delta \cdot \mathbb{P}(S = s)}$$

$$< \mathbb{P}(Y = 1|T = t),$$

as well as

$$\mathbb{P}_{new}(Y = 1|T = t_2)$$

$$= \sum_{s' \neq s} \mathbb{P}(Y = 1|S = s', T = t) \cdot \frac{\frac{1}{2}\mathbb{P}(T = t|S = s') \cdot \mathbb{P}(S = s')}{\frac{1}{2}\mathbb{P}(T = t) - \delta\mathbb{P}(S = s)}$$

$$= \mathbb{P}(Y = 1|T = t) \cdot \frac{\mathbb{P}(T = t)}{\mathbb{P}(T = t) - 2\delta \cdot \mathbb{P}(S = s)}$$

$$> \mathbb{P}(Y = 1|T = t).$$

Therefore, $\mathbb{P}_{new}(Y = 1|T)$ is a mean-preserving spread of $\mathbb{P}(Y = 1|T)$, and thus, the new $\mathbb{P}_{new}(Y, T)$ Blackwell dominates $\mathbb{P}(Y, T)$, but the reverse is not true. □

## C  PROOF OF LEMMA 4.3

PROOF. We first define sets $\mathcal{A}_t$, $\mathcal{B}_t$, and $C_t$ as follows. For each $t \in \widetilde{\mathcal{T}}$, define $\mathcal{A}_t$ to be the set of $s \in \mathcal{S}$ that has $\mathbb{P}(Y = 1|S = s, T = t) = 1$, and define $\mathcal{B}_t$ to be the set of $s \in \mathcal{A}_t$ that has $\mathbb{P}(T = t|S = s) = H_t$, and define $C_t$ to be the set of $s \notin \mathcal{A}_t$ that has $\mathbb{P}(T = t|S = s) = H_t$.

Similar to the definition of $\mathcal{A}_t$, $\mathcal{B}_t$, and $C_t$, we define $\mathcal{D}_s$ to be the set of $t \in \widetilde{\mathcal{T}}$ that has $\mathbb{P}(Y = 1|S = s, T = t) = 1$, and define $\mathcal{E}_s$ to be the set of $t \in \mathcal{D}_s$ that has $\mathbb{P}(T = t|S = s) = H_t$, and define $\mathcal{F}_s$ to be the set of $t \notin \mathcal{D}_s$ that has $\mathbb{P}(T = t|S = s) = H_t$.

We introduce 0-1 crossing blocks and H-L crossing blocks, and show that both types of crossing blocks cannot exist in a Blackwell optimal information structure.

DEFINITION C.1 (0-1 CROSSING BLOCKS). *A 0-1 crossing block is defined by $s_1, s_2 \in \mathcal{S}$ and $t_1, t_2 \in \widetilde{\mathcal{T}}$ with*

$$\mathbb{P}(Y = 1|s_1, t_1) = 0, \qquad \mathbb{P}(Y = 1|s_1, t_2) = 1,$$
$$\mathbb{P}(Y = 1|s_2, t_1) = 1, \qquad \mathbb{P}(Y = 1|s_2, t_2) = 0.$$

*In other words, we have $s_1 \notin \mathcal{A}_{t_1}$, $s_1 \in \mathcal{A}_{t_2}$, $s_2 \in \mathcal{A}_{t_1}$, and $s_2 \notin \mathcal{A}_{t_2}$.*

DEFINITION C.2 (H-L CROSSING BLOCKS). *An H-L crossing block is defined by $s_1, s_2 \in \mathcal{S}$ and $t_1, t_2 \in \widetilde{\mathcal{T}}$ with either (1) or (2),*

(1) *it holds that $s_1 \in \mathcal{B}_{t_1}$, $s_1 \in \mathcal{A}_{t_2} \setminus \mathcal{B}_{t_2}$, $s_2 \notin \mathcal{A}_{t_1}$, $s_2 \in \mathcal{A}_{t_2}$, in other words, we have*

$$\mathbb{P}(Y = 1|s_1, t_1) = 1, \quad \mathbb{P}(T = t_1|s_1) = H_{t_1},$$
$$\mathbb{P}(Y = 1|s_1, t_2) = 1, \quad \mathbb{P}(T = t_2|s_1) = L_{t_2},$$
$$\mathbb{P}(Y = 1|s_2, t_1) = 0, \quad \mathbb{P}(Y = 1|s_2, t_2) = 1.$$

(2) *it holds that $s_1 \in \overline{\mathcal{A}}_{t_1} \setminus C_{t_1}$, $s_1 \in C_{t_2}$, $s_2 \notin \mathcal{A}_{t_1}$, $s_2 \in \mathcal{A}_{t_2}$, in other words, we have*

$$\mathbb{P}(Y = 1|s_1, t_1) = 0, \quad \mathbb{P}(T = t_1|s_1) = L_{t_1},$$
$$\mathbb{P}(Y = 1|s_1, t_2) = 0, \quad \mathbb{P}(T = t_2|s_1) = H_{t_2},$$
$$\mathbb{P}(Y = 1|s_2, t_1) = 0, \quad \mathbb{P}(Y = 1|s_2, t_2) = 1.$$

LEMMA C.1. *A Blackwell optimal information structure must not have a 0-1 crossing block or an H-L crossing block.*

PROOF. We first prove that a "0-1 crossing block" cannot exist. For simplicity we write $p_{11} = \mathbb{P}(S = s_1, T = t_1)$ and $p_{12}, p_{21}, p_{22}$ similarly. Suppose a distribution $\mathbb{P}(Y, S, T)$ has a "0-1 crossing block". We show that we can slightly change $\mathbb{P}(Y, S, T)$ to $\widetilde{\mathbb{P}}(Y, S, T)$ so that $\widetilde{\mathbb{P}}(Y, T)$ Blackwell dominates $\mathbb{P}(Y, T)$, while preserving the marginal distribution $\mathbb{P}(Y, S)$ and $\mathbb{P}(S, T)$. We change the conditional distribution as follows

$$\widetilde{\mathbb{P}}(Y = 1|s_1, t_1) = \delta_1, \qquad \widetilde{\mathbb{P}}(Y = 1|s_1, t_2) = 1 - \frac{p_{11}}{p_{12}} \cdot \delta_1,$$
$$\widetilde{\mathbb{P}}(Y = 1|s_2, t_1) = 1 - \frac{p_{11}}{p_{21}} \cdot \delta_2, \qquad \widetilde{\mathbb{P}}(Y = 1|s_2, t_2) = \frac{p_{11}}{p_{22}} \cdot \delta_2,$$

where $\delta_1, \delta_2 \in (0, 1)$ will be determined later. We keep $\mathbb{P}(S, T)$ the same and it is not difficult to see that $\mathbb{P}(Y, S)$ is preserved. We set $\delta_1, \delta_2$ in a way that $\widetilde{\mathbb{P}}(Y = 1|T)$ is a mean-preserving spread of $\mathbb{P}(Y = 1|T)$, i.e., the posteriors after observing $T$ are only more

"extreme". By simple calculation, we have

$$\widetilde{\mathbb{P}}\left(Y=1, T=t_1\right)$$

$$= \sum_s \widetilde{\mathbb{P}}\left(Y=1, T=t_1, S=s\right)$$

$$= \widetilde{\mathbb{P}}\left(Y=1, T=t_1, S=s_1\right) + \widetilde{\mathbb{P}}\left(Y=1, T=t_1, S=s_2\right)$$

$$\quad + \mathbb{P}\left(Y=1, T=t_1, S \neq s_1, s_2\right)$$

$$= p_{21} + p_{11}(\delta_1 - \delta_2) + \mathbb{P}\left(Y=1, T=t_1, S \neq s_1, s_2\right)$$

$$= p_{11}(\delta_1 - \delta_2) + \mathbb{P}\left(Y=1, T=t_1\right).$$

The last equality holds because $\mathbb{P}\left(Y=1, T=t_1, S=s_1\right) = 0$ and $\mathbb{P}\left(Y=1, T=t_1, S=s_2\right) = p_{21}$. Therefore

$$\widetilde{\mathbb{P}}\left(Y=1 | T=t_1\right) = \frac{p_{11}}{\mathbb{P}\left(T=t_1\right)} \cdot (\delta_1 - \delta_2) + \mathbb{P}\left(Y=1 | T=t_1\right).$$

Similarly,

$$\widetilde{\mathbb{P}}\left(Y=1 | T=t_2\right) = -\frac{p_{11}}{\mathbb{P}\left(T=t_2\right)} \cdot (\delta_1 - \delta_2) + \mathbb{P}\left(Y=1 | T=t_2\right).$$

By our assumption that $\mathbb{P}\left(s, t\right) > 0$ for all $s, t$, we have $p_{11} > 0$ and $\mathbb{P}\left(Y=1 | T=t_1\right), \mathbb{P}\left(Y=1 | T=t_2\right) \in (0, 1)$. Then by choosing $\delta_1 > \delta_2$ when $\mathbb{P}\left(Y=1 | T=t_1\right) > \mathbb{P}\left(Y=1 | T=t_2\right)$ and $\delta_1 < \delta_2$ when $\mathbb{P}\left(Y=1 | T=t_1\right) \leq \mathbb{P}\left(Y=1 | T=t_2\right)$, we make the posteriors more extreme. The new distribution $\widetilde{P}$ preserves the inferential privacy constraint because the marginal distribution of $S$ and $T$ keeps the same, i.e., $\widetilde{\mathbb{P}}\left(S, T\right) = \mathbb{P}\left(S, T\right)$.

Similarly, we can prove that a "H-L crossing block" cannot exist either. Suppose $\mathbb{P}\left(Y, S, T\right)$ has a "H-L" crossing block of the first type. (The proof for the second type is entirely similar.) Let $p_1 = \mathbb{P}\left(S=s_1\right), p_{21} = \mathbb{P}\left(S=s_2, T=t_1\right)$, and $p_{22} = \mathbb{P}\left(S=s_2, T=t_2\right)$. We again slightly change $\mathbb{P}\left(Y, S, T\right)$ to $\widetilde{\mathbb{P}}\left(Y, S, T\right)$ by

$$\widetilde{\mathbb{P}}\left(T=t_1 | S=s_1\right) = \mathbb{P}\left(T=t_1 | S=s_1\right) - \frac{\delta_1}{p_1},$$

$$\widetilde{\mathbb{P}}\left(T=t_2 | S=s_1\right) = \mathbb{P}\left(T=t_2 | S=s_1\right) + \frac{\delta_1}{p_1},$$

$$\widetilde{\mathbb{P}}\left(Y=1 | s_2, t_1\right) = \frac{1}{p_{21}} \cdot \delta_2,$$

$$\widetilde{\mathbb{P}}\left(Y=1 | s_2, t_2\right) = 1 - \frac{1}{p_{22}} \cdot \delta_2.$$

Again, by simple calculation we have $\widetilde{\mathbb{P}}\left(Y=1 | T=t_1\right) = \frac{1}{\mathbb{P}(T=t_1)} \cdot (\delta_1 - \delta_2) + \mathbb{P}\left(Y=1 | T=t_1\right)$ and $\widetilde{\mathbb{P}}\left(Y=1 | T=t_2\right) = -\frac{1}{\mathbb{P}(T=t_2)} \cdot (\delta_1 - \delta_2) + \mathbb{P}\left(Y=1 | T=t_2\right)$. Therefore by choosing $\delta_1 > \delta_2$ when $\mathbb{P}\left(Y=1 | T=t_1\right) > \mathbb{P}\left(Y=1 | T=t_2\right)$ and $\delta_1 < \delta_2$ when $\mathbb{P}\left(Y=1 | T=t_1\right) \leq \mathbb{P}\left(Y=1 | T=t_2\right)$, we make the posteriors more extreme. The new distribution $\widetilde{P}$ will preserve the inferential privacy constraint when we choose small enough $\delta_1$ and $\delta_2$. Because (1) by the definition of "H-L crossing blocks" and $\mathcal{B}_t$, we have $\mathbb{P}\left(T=t_1 | S=s_1\right) = H_t$, $\mathbb{P}\left(T=t_2 | S=s_1\right) = L_t$, so by slightly decreasing $\mathbb{P}\left(T=t_1 | S=s_1\right) = H_t$ and increasing $\mathbb{P}\left(T=t_2 | S=s_1\right) = L_t$, the inferential privacy constraint is still satisfied; (2) we do not change the the marginal distribution of $\mathbb{P}\left(S, T\right)$ when $S=s_2$, i.e., $\widetilde{\mathbb{P}}\left(s_2, t_1\right) = \mathbb{P}\left(s_2, t_1\right)$ and $\widetilde{\mathbb{P}}\left(s_2, t_2\right) = \mathbb{P}\left(s_2, t_2\right)$.

$\square$

If an information structure $\mathbb{P}\left(Y, S, T\right)$ does not have a "0-1 crossing block", then for any pair of $t_1, t_2 \in \widetilde{\mathcal{T}}$, we either have $\mathcal{A}_{t_1} \subseteq \mathcal{A}_{t_2}$

or $\mathcal{A}_{t_2} \subseteq \mathcal{A}_{t_1}$. (Otherwise, an arbitrary $s_1 \in \mathcal{A}_{t_2} \setminus \mathcal{A}_{t_1}$ and an arbitrary $s_2 \in \mathcal{A}_{t_1} \setminus \mathcal{A}_{t_2}$ and $t_1, t_2$ will form a "0-1 crossing block".) Similarly, for any pair of $s_1, s_2 \in \mathcal{S}$, we either have $\mathcal{D}_{s_1} \subseteq \mathcal{D}_{s_2}$ or $\mathcal{D}_{s_2} \subseteq \mathcal{D}_{s_1}$.

Lemma C.2. *Consider any Blackwell optimal information structure* $\mathbb{P}\left(Y, S, T\right)$. *For any pair* $t_1, t_2 \in \widetilde{\mathcal{T}}$, *if* $\mathcal{A}_{t_1} \subset \mathcal{A}_{t_2}$, *then we must have* $\mathcal{B}_{t_1} \subseteq \mathcal{B}_{t_2}$ *and* $C_{t_2} \subseteq C_{t_1}$. *For any pair of* $s_1, s_2 \in \mathcal{S}$, *if* $\mathcal{D}_{s_1} \subset \mathcal{D}_{s_2}$, *then we must have* $\mathcal{E}_{s_1} \subseteq \mathcal{E}_{s_2}$ *and* $\mathcal{F}_{s_2} \subseteq \mathcal{F}_{s_1}$.

Proof. We prove that for any pair $t_1, t_2 \in \widetilde{\mathcal{T}}$, if $\mathcal{A}_{t_1} \subset \mathcal{A}_{t_2}$, then we must have $\mathcal{B}_{t_1} \subseteq \mathcal{B}_{t_2}$ and $C_{t_2} \subseteq C_{t_1}$. The proof for $\mathcal{D}, \mathcal{E}, \mathcal{F}$ is entirely similar.

Consider any $t_1, t_2 \in \widetilde{\mathcal{T}}$ with $\mathcal{A}_{t_1} \subset \mathcal{A}_{t_2}$. We first prove that we must have $\mathcal{B}_{t_1} \subseteq \mathcal{B}_{t_2}$. Suppose to the contrary, $\mathcal{B}_{t_1}$ is not a subset of $\mathcal{B}_{t_2}$, we claim that there must exist an "H-L crossing" block. We first find a valid $s_1$. Since $\mathcal{B}_{t_1}$ is not a subset of $\mathcal{B}_{t_2}$, we can find $s_1$ with $s_1 \in \mathcal{B}_{t_1}$ and $s_1 \notin \mathcal{B}_{t_2}$. By definition, $\mathcal{B}_{t_1} \subseteq \mathcal{A}_{t_1}$, and by assumption, $\mathcal{A}_{t_1} \subset \mathcal{A}_{t_2}$, therefore it is guaranteed that $s_1 \in \mathcal{B}_{t_1} \subseteq \mathcal{A}_{t_1} \subset \mathcal{A}_{t_2}$. So we find an $s_1$ with $s_1 \in \mathcal{B}_{t_1}$ and $s_1 \in \mathcal{A}_{t_2} \setminus \mathcal{B}_{t_2}$. We then find a valid $s_2$. Since $\mathcal{A}_{t_1} \subset \mathcal{A}_{t_2}$, we can find $s_2$ with $s_2 \in \mathcal{A}_{t_2}$ and $s_2 \notin \mathcal{A}_{t_1}$. Then by definition, $s_1, s_2, t_1, t_2$ form an "H-L crossing" block. Therefore, we must have $\mathcal{B}_{t_1} \subseteq \mathcal{B}_{t_2}$.

Next, we show that for any $t_1, t_2 \in \widetilde{\mathcal{T}}$ with $\mathcal{A}_{t_1} \subset \mathcal{A}_{t_2}$, we must have $C_{t_2} \subseteq C_{t_1}$. Suppose to the contrary, $C_{t_2}$ is not a subset of $C_{t_1}$, we claim that there must exist an "H-L crossing" block. We first find a valid $s_1$. Since $C_{t_2}$ is not a subset of $C_{t_1}$, we can find $s_1$ with $s_1 \in C_{t_2}$ and $s_1 \notin C_{t_1}$. By our definition and our assumption, $C_{t_2} \subseteq \overline{\mathcal{A}}_{t_2} \subset \overline{\mathcal{A}}_{t_1}$, therefore it is guaranteed that $s_1 \in C_{t_2} \subseteq \overline{\mathcal{A}}_{t_2} \subset \overline{\mathcal{A}}_{t_1}$. So we find an $s_1$ with $s_1 \in C_{t_2}$ and $s_1 \in \overline{\mathcal{A}}_{t_1} \setminus C_{t_1}$. We then find a valid $s_2$. Because $\mathcal{A}_{t_1} \subset \mathcal{A}_{t_2}$, we can find $s_2$ with $s_2 \in \mathcal{A}_{t_2}$ and $s_2 \notin \mathcal{A}_{t_1}$. Then by definition, $s_1, s_2, t_1, t_2$ form an "H-L crossing" block. Therefore we must have $C_{t_2} \subseteq C_{t_1}$. $\square$

Now we are ready to prove Lemma 4.3. Suppose we sort $s \in \mathcal{S}$ from largest $\mathcal{D}_s$ to smallest $\mathcal{D}_s$, and sort $t \in \widetilde{\mathcal{T}}$ from largest $\mathcal{A}_t$ to smallest $\mathcal{A}_t$. Then we must have the region of $\mathcal{A}_t$ being an upper-left region. Otherwise, it will conflict the ordering $\mathcal{A}_{t_k} \subseteq \mathcal{A}_{t_j}$ or $\mathcal{D}_{s_k} \subseteq \mathcal{D}_{s_j}$. According to Lemma C.2, we have $\mathcal{B}_{t_k} \subseteq \mathcal{B}_{t_j}$ and $\mathcal{E}_{s_k} \subseteq \mathcal{E}_{s_j}$ for any $j \leq k$. Therefore region $\mathcal{B}$ must be upper-left as well. In addition, according to Lemma C.2, we must have $C_{t_k} \subseteq C_{t_j}$ and $\mathcal{E}_{s_k} \subseteq \mathcal{E}_{s_j}$ for any $j \geq k$. Then for the same reason, region $C$ must be lower-right. We thus prove the lemma. $\square$

# D PROOF OF THEOREM 4.1

Proof. Suppose $s_1, \ldots, s_n \in \mathcal{S}$ are ordered in decreasing order of $\mathbb{P}\left(Y=1 | S=s\right)$. For any $\varepsilon$-inferentially-private Blackwell optimal information structure $\hat{\mathbb{P}}\left(Y, S, \hat{T}\right)$, suppose there are $k$ unique values of $\hat{\mathbb{P}}\left(Y=1 | \hat{T}\right)$, denoted as $v_1, \ldots, v_k$. Suppose $v_1 > \ldots > v_k$ without loss of generality, and denote $\hat{\mathcal{T}}_i = \left\{\hat{t} \in \widetilde{\mathcal{T}} : \hat{\mathbb{P}}\left(Y=1 | \hat{T}=\hat{t}\right) = v_i\right\}$ where $i \in [k]$. Based on Lemmas 4.2 and 4.3, we can get that

$$\forall s \in \mathcal{S}, i \in [k], \hat{t}_1, \hat{t}_2 \in \hat{\mathcal{T}}_i : \hat{\mathbb{P}}\left(Y=1 | S=s, \hat{T}=\hat{t}_1\right) = \hat{\mathbb{P}}\left(Y=1 | S=s, \hat{T}=\hat{t}_2\right),$$

$$\forall s \in \mathcal{S}, i \in [k], \hat{t}_1, \hat{t}_2 \in \hat{\mathcal{T}}_i : \hat{\mathbb{P}}\left(S=s | \hat{T}=\hat{t}_1\right) = \hat{\mathbb{P}}\left(S=s | \hat{T}=\hat{t}_2\right).$$

We can construct an information structure $\mathbb{P}(S, Y, T)$ such that

$$\forall s \in \mathcal{S}: \quad \mathbb{P}(T = t_i | S = s) = \sum_{\hat{t} \in \hat{\mathcal{T}}_i} \hat{\mathbb{P}}\left(\hat{T} = \hat{t} | S = s\right).$$

With simple calculations, we can get that $\mathbb{P}(T = t_i) = \sum_{\hat{t} \in \hat{\mathcal{T}}_i} \hat{\mathbb{P}}\left(\hat{T} = \hat{t}\right)$, $\mathbb{P}(Y = 1 | T = t_i) = v_i$, and $\mathbb{P}(S = s | T = t_i) = \hat{\mathbb{P}}\left(S = s | \hat{T} = \hat{t}\right), \forall i \in [k], s \in \mathcal{S}, \hat{t} \in \hat{\mathcal{T}}_i$. Therefore, $\mathbb{P}(S, Y, T)$ is an equivalent information structure to $\hat{\mathbb{P}}\left(S, Y, \hat{T}\right)$, and $t_1, \ldots, t_k \in \mathcal{T}$ are ordered in decreasing order of $q_t = \mathbb{P}(Y = 1 | T = t)$.

Intuitively, $\mathbb{P}(S, Y, T)$ is a compressed version of $\hat{\mathbb{P}}\left(S, Y, \hat{T}\right)$, by merging the columns that share the same geometric characterization. As illustrated in Fig. 9, the columns corresponding to $\hat{t}_3$ and $\hat{t}_4$ in $\hat{\mathbb{P}}\left(S, Y, \hat{T}\right)$ shares the same geometric pattern, and thus can be merged as a single column (corresponding to $t_3$) in $\mathbb{P}(S, Y, T)$.

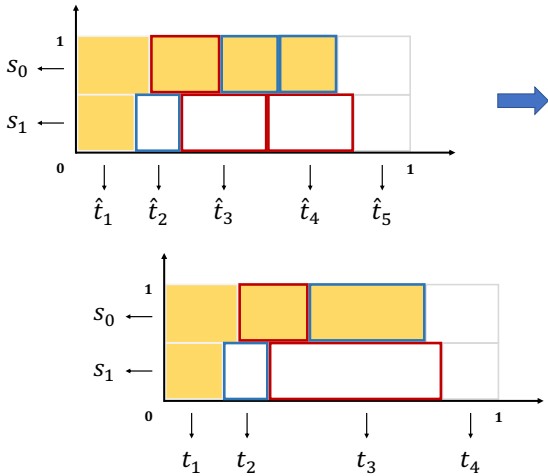

**Figure 9: Information structure of $\hat{\mathbb{P}}\left(S, Y, \hat{T}\right)$ (left) and $\mathbb{P}(S, Y, T)$ (right) with binary secret $S \in \{s_0, s_1\}$. In $\hat{\mathbb{P}}\left(S, Y, \hat{T}\right)$, the columns corresponding to $\hat{t}_3$ and $\hat{t}_4$ has the same geometric pattern. In $\mathbb{P}(S, Y, T)$, those columns are merged as a single column (the column corresponding to $t_3$).**

For any convex utility function $u$, we have

$$\mathbb{E}_t[u(q_t)] = \sum_{i \in [k]} \mathbb{P}(T = t_i) u\left(q_{t_i}\right)$$

$$= \sum_{i \in [k]} \sum_{\hat{t} \in \hat{\mathcal{T}}_i} \hat{\mathbb{P}}\left(\hat{T} = \hat{t}\right) u(v_i)$$

$$= \sum_{\hat{t} \in \hat{\mathcal{T}}} \hat{\mathbb{P}}\left(\hat{T} = \hat{t}\right) u\left(q_{\hat{t}}\right)$$

$$= \mathbb{E}_{\hat{t}}\left[u\left(q_{\hat{t}}\right)\right].$$

Therefore, from Thm. 2.1, we know that $\mathbb{P}(S, Y, T)$ is an optimal information structure equivalent to $\hat{\mathbb{P}}\left(S, Y, \hat{T}\right)$.

For each $t \in \widetilde{\mathcal{T}}$, define $\mathcal{A}_t$ to be the set of $s \in \mathcal{S}$ that has $\mathbb{P}(Y = 1 | S = s, T = t) = 1$, and define $\mathcal{B}_t$ to be the set of $s \in \mathcal{A}_t$ that has $\mathbb{P}(T = t | S = s) = H_t$, and define $\mathcal{C}_t$ to be the set of $s \notin \mathcal{A}_t$ that has $\mathbb{P}(T = t | S = s) = H_t$. From Lemma 4.3, we know that for any $i \in \{2, \cdots, k - 1\}$, we have $\mathcal{A}_{t_i} = \{s_1, \ldots, s_{a_i}\}$ and $\mathcal{B}_{t_i} = \{s_1, \ldots, s_{b_i}\}$ and $\mathcal{C}_{t_i} = \{s_{c_i}, \ldots, s_n\}$ for some $a_i, b_i, c_i$ with

$$1 \leq a_i < n, \quad 0 \leq b_i < n, \quad 1 < c_i \leq n + 1,$$
$$a_{i+1} \leq a_i \leq a_{i-1}, \quad b_{i+1} \leq b_i \leq b_{i-1}, \quad c_{i+1} \leq c_i \leq c_{i-1},$$

where $\mathcal{B}_{t_i} = \emptyset$ when $b_i = 0$ and $\mathcal{C}_{t_i} = \emptyset$ when $c_i = n + 1$. Since $(a_i, b_i, c_i)$ and $(a_{i+1}, b_{i+1}, c_{i+1})$ differ for at least one element in $\mathbb{P}(S, Y, T)$, we have $k - 2 \leq n - 1 + n + n = 3n - 1$, i.e., $|\mathcal{T}| = k \leq 3n + 1 = 3|\mathcal{S}| + 1$. □

# E PROOF OF LEMMA 5.1

Proof. Without loss of generality, we let $\mathbb{P}(Y = 1 | S = s_1) \leq \mathbb{P}(Y = 1 | S = s_0)$. From the proof of Thm. 4.1, we know that there is an $\varepsilon$-inferentially-private Blackwell optimal information structure $\hat{\mathbb{P}}\left(S, Y, \hat{T}\right)$ where there is only one output, denoted as $t_1$ and $t_k$, that satisfies $\hat{\mathbb{P}}\left(Y = 1 | \hat{T} = \hat{t}_1\right) = 1$ and $\hat{\mathbb{P}}\left(Y = 1 | \hat{T} = \hat{t}_k\right) = 0$ respectively. Then we can get that $\hat{\mathbb{P}}\left(Y = 1 | S = s_1, \hat{T} = \hat{t}\right) = 0, \forall \hat{t} \neq \hat{t}_1$ and $\hat{\mathbb{P}}\left(Y = 1 | S = s_0, \hat{T} = \hat{t}\right) = 1, \forall \hat{t} \neq \hat{t}_k$, because otherwise, based on Lemma 4.3, we have $\hat{\mathbb{P}}\left(Y = 1 | \hat{T} = \hat{t}\right) = 1, \exists \hat{t} \neq \hat{t}_1$ or $\hat{\mathbb{P}}\left(Y = 1 | \hat{T} = \hat{t}\right) = 0, \exists \hat{t} \neq \hat{t}_{n+1}$. With simple calculation, we can get that $\hat{\mathbb{P}}\left(\hat{T} = \hat{t}_1 | S = s_1\right) = q_{s_1}$ and $\hat{\mathbb{P}}\left(\hat{T} = \hat{t}_1 | S = s_0\right) = 1 - q_{s_0}$. Based on Lemmas 4.2 and 4.3, we can get that $\exists 1 \leq i_0 \leq k$:

$$\hat{\mathbb{P}}\left(\hat{T} = \hat{t}_i | S = s_0\right) = \begin{cases} H_{\hat{t}_i}, & 1 < i \leq i_0 \\ L_{\hat{t}_i}, & i_0 < i < k \end{cases},$$

$$\hat{\mathbb{P}}\left(\hat{T} = \hat{t}_i | S = s_1\right) = \begin{cases} H_{\hat{t}_i}, & i_0 < i < k \\ L_{\hat{t}_i}, & 1 < i \leq i_0 \end{cases}.$$

We can construct an information structure $\mathbb{P}(S, Y, T)$ where $\forall j \in \{0, 1\}$:

$$l_1^{(j)} = \mathbb{P}(T = t_1 | S = s_j) = \hat{\mathbb{P}}\left(\hat{T} = \hat{t}_1 | S = s_j\right),$$

$$l_2^{(j)} = \mathbb{P}(T = t_2 | S = s_j) = \sum_{1 < i \leq i_0} \hat{\mathbb{P}}\left(\hat{T} = \hat{t}_i | S = s_j\right),$$

$$l_3^{(j)} = \mathbb{P}(T = t_3 | S = s_j) = \sum_{i_0 < i < k} \hat{\mathbb{P}}\left(\hat{T} = \hat{t}_i | S = s_j\right),$$

$$l_4^{(j)} = \mathbb{P}(T = t_4 | S = s_j) = \hat{\mathbb{P}}\left(\hat{T} = \hat{t}_k | S = s_j\right).$$

With similar analysis of the proof of Thm. 4.1, we can verify that $\mathbb{P}(S, Y, T)$ is an optimal information structure equivalent to $\hat{\mathbb{P}}\left(S, Y, \hat{T}\right)$.

We have $l_1^{(1)} = \hat{\mathbb{P}}\left(\hat{T} = \hat{t}_1 | S = s_j\right) = q_{s_1}$ and $l_4^{(0)} = \hat{\mathbb{P}}\left(\hat{T} = \hat{t}_k | S = s_j\right) = 1 - q_{s_0}$. Following the inferential privacy constraints, we have

$l_1^{(0)}/l_1^{(1)} \in [e^{-\varepsilon}, e^{\varepsilon}]$, $l_4^{(0)}/l_4^{(1)} \in [e^{-\varepsilon}, e^{\varepsilon}]$. Besides, we can get that

$$\frac{l_2^{(0)}}{l_2^{(1)}} = \frac{\sum_{1 < i \le i_0} \hat{\mathbb{P}}(T = \hat{t}_i | S = s_0)}{\sum_{1 < i \le i_0} \hat{\mathbb{P}}(T = \hat{t}_i | S = s_1)} = \frac{\sum_{1 < i \le i_0} H_{\hat{t}_i}}{\sum_{1 < i \le i_0} L_{\hat{t}_i}} = e^{\varepsilon},$$

$$\frac{l_3^{(1)}}{l_3^{(0)}} = \frac{\sum_{i_0 < i < k} \hat{\mathbb{P}}(T = \hat{t}_i | S = s_1)}{\sum_{i_0 < i < k} \hat{\mathbb{P}}(T = \hat{t}_i | S = s_0)} = \frac{\sum_{i_0 < i < k} H_{\hat{t}_i}}{\sum_{i_0 < i < k} L_{\hat{t}_i}} = e^{\varepsilon}.$$

Finally, we have $\forall j \in \{0, 1\} : \sum_{i \in [4]} l_i^{(j)} = \sum_{i \in [4]} \mathbb{P}(T = t_i | S = s_j) = 1$. □

## F PROOF OF LEMMA 5.2

PROOF. We first prove that there exists a unique feasible dominant point. According to Lemma 5.1, we have $l_1^{(1)} = q_{s_1}, l_4^{(0)} = 1 - q_{s_0}, l_2^{(0)} = e^{\varepsilon} l_2^{(1)}, l_3^{(1)} = e^{\varepsilon} l_3^{(0)}, \sum_{i \in [4]} l_i^{(0)} = \sum_{i \in [4]} l_i^{(1)} = 1$. Therefore, we can represent $l_2^{(1)}, l_3^{(1)}$ by $l_1^{(0)}$ and $l_4^{(1)}$ based on the following two equations:

$$q_{s_1} + l_2^{(1)} + l_3^{(1)} + l_4^{(1)} = 1,$$

$$l_1^{(0)} + e^{\varepsilon} l_2^{(1)} + e^{-\varepsilon} l_3^{(1)} + 1 - q_{s_0} = 1.$$

We can get that

$$l_2^{(1)} = \frac{-e^{\varepsilon} l_1^{(0)} + l_4^{(1)} + e^{\varepsilon} q_{s_0} + q_{s_1} - 1}{e^{2\varepsilon} - 1}, \tag{10}$$

$$l_3^{(1)} = e^{\varepsilon} \cdot \frac{l_1^{(0)} - e^{\varepsilon} l_4^{(1)} - q_{s_0} - e^{\varepsilon} q_{s_1} + e^{\varepsilon}}{e^{2\varepsilon} - 1}. \tag{11}$$

Based on the constraints that $\forall i \in [4], j \in [2] : l_i^{(j)} \ge 0$, $l_1^{(0)} \in [e^{-\varepsilon} l_1^{(1)}, e^{\varepsilon} l_1^{(1)}]$, and $l_4^{(1)} \in [e^{-\varepsilon} l_4^{(0)}, e^{\varepsilon} l_4^{(0)}]$, we have

$$- e^{\varepsilon} l_1^{(0)} + l_4^{(1)} + e^{\varepsilon} q_{s_0} + q_{s_1} - 1 \ge 0,$$

$$l_1^{(0)} - e^{\varepsilon} l_4^{(1)} - q_{s_0} - e^{\varepsilon} q_{s_1} + e^{\varepsilon} \ge 0,$$

$$e^{\varepsilon} q_{s_1} \ge l_1^{(0)} \ge e^{-\varepsilon} q_{s_1},$$

$$e^{\varepsilon} (1 - q_{s_0}) \ge l_4^{(1)} \ge e^{-\varepsilon} (1 - q_{s_0}).$$

Let $R_1 = \frac{q_{s_0}}{q_{s_1}}$, and $R_2 = \frac{1 - q_{s_1}}{1 - q_{s_0}}$. Then we can get that

$$l_1^{(0)} \le \begin{cases} q_{s_0}, & R_1 \le e^{\varepsilon} \cap R_2 \le e^{\varepsilon} \\ e^{\varepsilon} q_{s_1}, & R_1 > e^{\varepsilon} \cap (R_2 \le e^{\varepsilon} \cup (R_2 > e^{\varepsilon} \cap q_{s_1} < \frac{1}{1+e^{\varepsilon}})) \\ 1 - e^{-\varepsilon}(1 - q_{s_1}), & R_2 > e^{\varepsilon} \cap (R_1 \le e^{\varepsilon} \cup (R_1 > e^{\varepsilon} \cap q_{s_1} \ge \frac{1}{1+e^{\varepsilon}})) \end{cases},$$

$$l_4^{(1)} \le \begin{cases} 1 - q_{s_1}, & R_1 \le e^{\varepsilon} \cap R_2 \le e^{\varepsilon} \\ 1 - e^{-\varepsilon} q_{s_0}, & R_1 > e^{\varepsilon} \cap (R_2 \le e^{\varepsilon} \cup (R_2 > e^{\varepsilon} \cap q_{s_0} \le \frac{1}{1+e^{-\varepsilon}})) \\ e^{\varepsilon}(1 - q_{s_0}), & R_2 > e^{\varepsilon} \cap (R_1 \le e^{\varepsilon} \cup (R_1 > e^{\varepsilon} \cap q_{s_0} > \frac{1}{1+e^{-\varepsilon}})) \end{cases} \tag{12}$$

and the upper bounds of $l_1^{(0)}$ and $l_4^{(1)}$ can be achieved simultaneously, i.e., there exists a unique feasible dominant point.

To prove the feasible dominant point universally maximizing the objective function, we first prove the following two inequalities for any convex utility function $u$:

$$p_{s_1} u(q_{t_4}) + \frac{p_{s_1} + e^{\varepsilon} p_{s_0}}{e^{2\varepsilon} - 1} u(q_{t_2}) \ge \frac{e^{2\varepsilon} p_{s_1} + e^{\varepsilon} p_{s_0}}{e^{2\varepsilon} - 1} u(q_{t_3}), \tag{13}$$

$$p_{s_0} u(q_{t_1}) + \frac{e^{\varepsilon} p_{s_1} + p_{s_0}}{e^{2\varepsilon} - 1} u(q_{t_3}) \ge \frac{e^{2\varepsilon} p_{s_0} + e^{\varepsilon} p_{s_1}}{e^{2\varepsilon} - 1} u(q_{t_2}). \tag{14}$$

Since we have $q_{t_1} = 1, q_{t_4} = 0, q_{t_2} = \frac{l_2^{(0)} p_{s_0}}{l_2^{(0)} p_{s_0} + l_2^{(1)} p_{s_1}} = \frac{e^{\varepsilon} p_{s_0}}{e^{\varepsilon} p_{s_0} + p_{s_1}}$, $q_{t_3} = \frac{l_3^{(0)} p_{s_0}}{l_3^{(0)} p_{s_0} + l_3^{(1)} p_{s_1}} = \frac{p_{s_0}}{p_{s_0} + e^{\varepsilon} p_{s_1}}$, we can get that

$$p_{s_1} q_{t_4} + \frac{p_{s_1} + e^{\varepsilon} p_{s_0}}{e^{2\varepsilon} - 1} q_{t_2} = \frac{e^{\varepsilon} p_{s_0}}{e^{2\varepsilon} - 1} = \frac{e^{2\varepsilon} p_{s_1} + e^{\varepsilon} p_{s_0}}{e^{2\varepsilon} - 1} q_{t_3},$$

$$p_{s_0} q_{t_1} + \frac{e^{\varepsilon} p_{s_1} + p_{s_0}}{e^{2\varepsilon} - 1} q_{t_3} = \frac{e^{2\varepsilon} p_{s_0}}{e^{2\varepsilon} - 1} = \frac{e^{2\varepsilon} p_{s_0} + e^{\varepsilon} p_{s_1}}{e^{2\varepsilon} - 1} q_{t_2}.$$

Since $u$ is a convex function, we can easily get that Eqs. (13) and (14) hold based on Jensen's inequality.

Then we prove that if we fix the value of $l_1^{(0)}$ and reduce $l_4^{(1)}$, the value of objective function decreases. For a feasible set of values $\{l_i^{(j)}\}_{i \in [4], j \in [2]}$, where $l_2^{(1)} > 0$, let $U$ be the value of objective function, i.e., $U = \sum_{i \in [4]} p_{t_i} \cdot u(q_{t_i})$. Suppose we fix $l_1^{(0)}$ and decrease $l_4^{(1)}$ to $\tilde{l}_4^{(1)} = l_4^{(1)} - \Delta l$, then based on Eqs. (10) and (11), we have $\tilde{l}_2^{(1)} = l_2^{(1)} - \frac{\Delta l}{e^{2\varepsilon} - 1}, \tilde{l}_3^{(1)} = l_3^{(1)} + \frac{e^{2\varepsilon} \Delta l}{e^{2\varepsilon} - 1}$. Therefore, based on Eq. (13), we have

$$\tilde{U} = \sum_{i \in [4]} \tilde{P}_{t_i} \cdot u(q_{t_i})$$

$$= \sum_{i \in [4]} p_{t_i} \cdot u(q_{t_i}) + \sum_{j \in \{2,3,4\}} \left[ \left( \tilde{l}_j^{(0)} - l_j^{(0)} \right) p_{s_0} + \left( \tilde{l}_j^{(1)} - l_j^{(1)} \right) p_{s_1} \right] u(q_{t_j})$$

$$= U + \frac{e^{2\varepsilon} p_{s_1} + e^{\varepsilon} p_{s_0}}{e^{2\varepsilon} - 1} u(q_{t_3}) - \frac{p_{s_1} + e^{\varepsilon} p_{s_0}}{e^{2\varepsilon} - 1} u(q_{t_2}) - p_{s_1} u(q_{t_4})$$

$$\le U.$$

Similarly, based on Eq. (14), we can prove that if we fix the value of $l_4^{(1)}$ and reduce $l_1^{(0)}$, the value of objective function decreases.

Above all, we know that a necessary condition of the objective function $\mathbb{E}_t[u(q_t)]$ being maximized is that $l_1^{(0)}$ and $l_4^{(1)}$ are on the Pareto frontier maximizing each of these quantities individually. □

## G PROOF OF THEOREM 5.1

PROOF. When $l_1^{(0)}$ and $l_4^{(1)}$ achieve their feasible maximal values simultaneously, based on Eq. (12) and feasibility conditions in Lemma 5.1, we can easily get that

- When $R_1 \le e^{\varepsilon}, R_2 \le e^{\varepsilon}$:
  $l_4^{(1)} = 1 - q_{s_1}, \quad l_1^{(0)} = q_{s_0}, \quad l_2^{(1)} = l_3^{(1)} = 0.$
- When $R_1 \le e^{\varepsilon}, R_2 > e^{\varepsilon}$ or $R_1 > e^{\varepsilon}, R_2 > e^{\varepsilon}, q_{s_1} \ge \frac{1}{1+e^{\varepsilon}}$:
  $l_4^{(1)} = e^{\varepsilon}(1 - q_{s_0}), \quad l_1^{(0)} = 1 - e^{-\varepsilon}(1 - q_{s_1}), \quad l_2^{(1)} = 0, \quad l_3^{(1)} = 1 - q_{s_1} - e^{\varepsilon}(1 - q_{s_0}).$
- When $R_1 > e^{\varepsilon}, R_2 \le e^{\varepsilon}$ or $R_1 > e^{\varepsilon}, R_2 > e^{\varepsilon}, q_{s_0} \le \frac{1}{1+e^{-\varepsilon}}$:
  $l_4^{(1)} = 1 - e^{-\varepsilon} q_{s_0}, \quad l_1^{(0)} = e^{\varepsilon} q_{s_1}, \quad l_2^{(1)} = e^{-\varepsilon} q_{s_0} - q_{s_1}, \quad l_3^{(1)} = 0.$
- When $R_1 > e^{\varepsilon}, R_2 > e^{\varepsilon}, q_{s_0} > \frac{1}{1+e^{-\varepsilon}}, q_{s_1} < \frac{1}{1+e^{\varepsilon}}$:
  $l_4^{(1)} = e^{\varepsilon}(1 - q_{s_0}), \quad l_1^{(0)} = e^{\varepsilon} q_{s_1}, \quad l_2^{(1)} = \frac{1}{e^{\varepsilon}+1} - q_{s_1}, \quad l_3^{(1)} = e^{\varepsilon} q_{s_0} - \frac{e^{2\varepsilon}}{e^{\varepsilon}+1}.$

From Lemma 5.1, we know that this Blackwell optimal information structure is unique up to equivalence. □

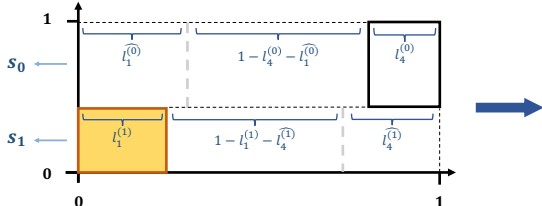

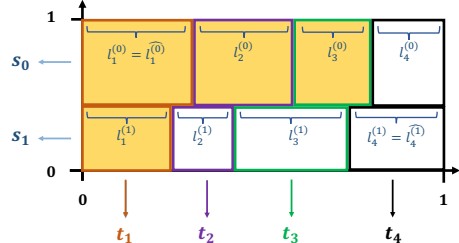

**Figure 10: When $R_1, R_2 > e^\varepsilon$ and $R = \frac{1-l_1^{(1)}-\widehat{l_4^{(1)}}}{1-l_4^{(0)}-l_1^{(0)}} \in (e^{-\varepsilon}, e^\varepsilon)$ (i.e., $q_{s_0} > \frac{1}{1+e^{-\varepsilon}}$ and $q_{s_1} < \frac{1}{1+e^\varepsilon}$), both $l_1^{(0)}$ and $l_4^{(1)}$ can achieve their upper bounds $\widehat{l_1^{(0)}}$ and $\widehat{l_4^{(1)}}$, and we can get feasible positive values of $l_2^{(1)}$ and $l_3^{(1)}$ based on Lemma 5.1.**

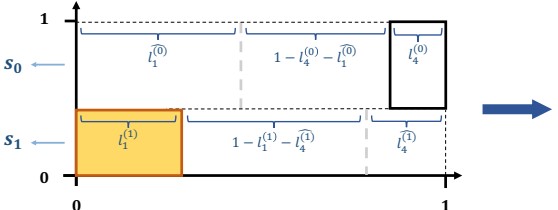

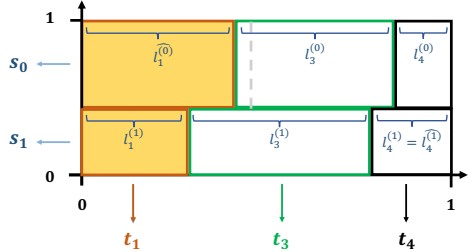

**Figure 11: When $R_1, R_2 > e^\varepsilon$ and $R = \frac{1-l_1^{(1)}-\widehat{l_4^{(1)}}}{1-l_4^{(0)}-l_1^{(0)}} \geq e^\varepsilon$ (i.e., $q_{s_1} \geq \frac{1}{1+e^\varepsilon}$), $l_4^{(1)}$ can achieve its upper bound $\widehat{l_4^{(1)}}$. To ensure $l_1^{(0)}$ to achieve its feasible maximal value, $l_2^{(0)}$ and $l_2^{(1)}$ should be set as 0.**

We give an intuitive explanation of the optimal information structure under the case where $R_1, R_2 > e^\varepsilon$, i.e, the inferential privacy property does not hold in the original secret-state structure $\mathbb{P}(S, Y)$. Let $\widehat{l_1^{(0)}} = e^\varepsilon l_1^{(1)}$ and $\widehat{l_4^{(1)}} = e^\varepsilon l_4^{(0)}$. From Lemma 5.1, we know that $l_1^{(0)} \leq \widehat{l_1^{(0)}}$ and $l_4^{(1)} \leq \widehat{l_4^{(1)}}$, and denote $R$ as $R = \frac{1-l_1^{(1)}-\widehat{l_4^{(1)}}}{1-l_4^{(0)}-l_1^{(0)}}$. When $q_{s_0} > \frac{1}{1+e^{-\varepsilon}}$ and $q_{s_1} < \frac{1}{1+e^\varepsilon}$, we have $R \in (e^{-\varepsilon}, e^\varepsilon)$, and both $l_1^{(0)}$ and $l_4^{(1)}$ can achieve their upper bounds $\widehat{l_1^{(0)}}$ and $\widehat{l_4^{(1)}}$. This is because in this case, $R = \frac{l_2^{(1)}+l_3^{(1)}}{l_2^{(0)}+l_3^{(0)}} \in (e^{-\varepsilon}, e^\varepsilon)$, and combining with Eq. (6), we can get feasible positive values of $l_2^{(1)}$ and $l_3^{(1)}$. When $q_{s_0} \leq \frac{1}{1+e^{-\varepsilon}}$ or $q_{s_1} \geq \frac{1}{1+e^\varepsilon}$, i.e., when $R \leq e^{-\varepsilon}$ or $R \geq e^\varepsilon$, to ensure both $l_1^{(0)}$ and $l_4^{(1)}$ reach their feasible maximal values, we have $\frac{1-l_1^{(1)}-l_4^{(1)}}{1-l_4^{(0)}-l_1^{(0)}} = \frac{l_2^{(1)}+l_3^{(1)}}{l_2^{(0)}+l_3^{(0)}} \in \{e^{-\varepsilon}, e^\varepsilon\}$, and thus $l_2^{(1)}$ or $l_3^{(1)}$ should be set as 0. The illustrations in Figs. 10 and 11 show the cases where $R \in (e^{-\varepsilon}, e^\varepsilon)$ and $R \geq e^\varepsilon$.

Furthermore, under different $\mathbb{P}(S, Y)$, we illustrate the $\varepsilon$-inferentially-private Blackwell optimal information structure in Fig. 12. With different values of $R_1, R_2, q_{s_0}, q_{s_1}$, the output signal set $\mathcal{T}$ varies.

## H  PROOF OF PROPOSITION 5.1

PROOF. Consider a utility function

$$u(q_t) = \begin{cases} \frac{L}{2} - Lq_t, & q_t \leq \frac{1}{2} \\ -\frac{L}{2} + Lq_t, & q_t > \frac{1}{2} \end{cases}$$

and a joint distribution $\mathbb{P}(S, Y)$ where $\mathbb{P}(S = s_0) = \mathbb{P}(S = s_1) = \frac{1}{2}$, $q_{s_0} = \frac{e^\varepsilon}{1+e^\varepsilon}, q_{s_1} = \frac{1}{1+e^\varepsilon}$. From Corollary 5.1, we know that the optimal mechanism contains two output signals $t_1, t_4$, and satisfies $p_{t_1} = p_{t_4} = \frac{1}{2}, q_{t_1} = 1, q_{t_4} = 0$. Then we can get that $U_\varepsilon = \frac{1}{2}u(0) + \frac{1}{2}u(1) = \frac{L}{2}$. Under the perfect privacy constraint, from Thm. 3.1, we know that the optimal mechanism contains three outputs $t_1', t_2', t_3'$, where $p_{t_1} = p_{t_3} = \frac{1}{1+e^\varepsilon}, p_{t_2} = \frac{e^\varepsilon-1}{1+e^\varepsilon}, q_{t_1} = 1, q_{t_3} = 0, q_{t_2} = \frac{1}{2}$. Then we can get that $U_0 = \frac{1}{1+e^\varepsilon}(u(0) + u(1)) + \frac{e^\varepsilon-1}{1+e^\varepsilon}u\left(\frac{1}{2}\right) = \frac{L}{1+e^\varepsilon}$. Let $L = 3\Delta\left(1 + \frac{2}{e^\varepsilon-1}\right)$, we have $U_\varepsilon - U_0 = \frac{L(e^\varepsilon-1)}{2(1+e^\varepsilon)} \geq \Delta$. □

## I  MECHANISM DESIGN FOR $n > 2$ SECRETS

In this section, we focus on general secrets with $n$ ($n > 2$) possible values, i.e., $\mathcal{S} = \{s_1, \ldots, s_n\}$. The main result is to derive a set of programs that lead to an optimal solution for any given utility function in the downstream decision-making problem. The design of the programs depends on our geometric characterization, which ensures that each program is linear.

We first capture the inferentially-private Blackwell optimal information structure as follows. Without loss of generality, we suppose $s_1, \ldots, s_n \in \mathcal{S}$ are ordered in decreasing order of $q_s = \mathbb{P}(Y = 1|S = s)$. Let $q_t = \mathbb{P}(Y = 1|T = t), p_t = \mathbb{P}(T = t)$, and $l_i^{(j)} = \mathbb{P}(T = t_i|S = s_i)$. From Thm. 4.1 and its proof, we know that any $\varepsilon$-inferentially-private Blackwell optimal information structure has an equivalent structure $\mathbb{P}(S, Y, T)$ where the number of output signals is at most $3n + 1$, and there is only one output, denoted as $t_1$ and $t_{n+1}$, that satisfies $q_{t_1} = \mathbb{P}(Y = 1|T = t_1) = 1$ and

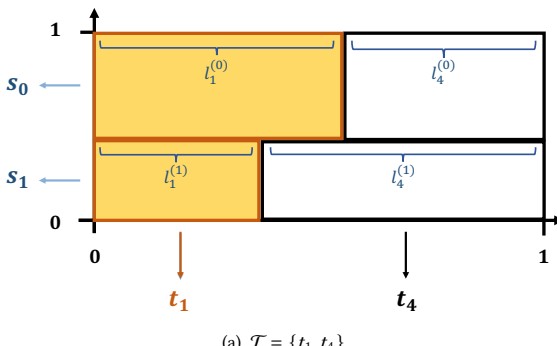

(a) $\mathcal{T} = \{t_1, t_4\}$

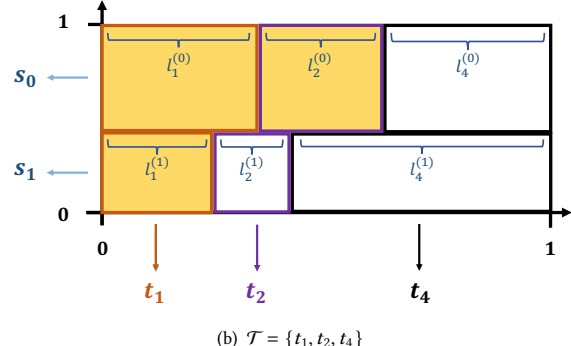

(b) $\mathcal{T} = \{t_1, t_2, t_4\}$

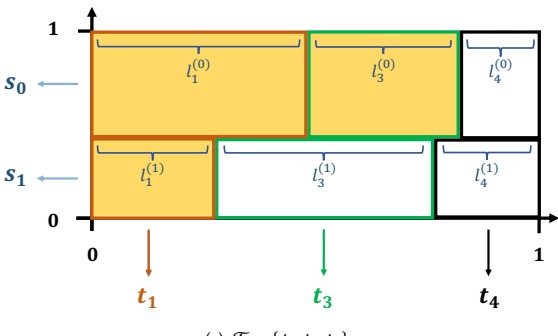

(c) $\mathcal{T} = \{t_1, t_3, t_4\}$

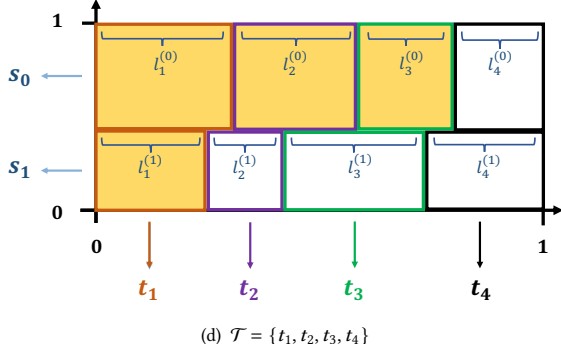

(d) $\mathcal{T} = \{t_1, t_2, t_3, t_4\}$

**Figure 12: For any $\mathbb{P}(S, Y)$, we can construct a unique $\varepsilon$-inferentially-private Blackwell optimal information structure $\mathbb{P}(S, Y, T)$, where the output signal set $\mathcal{T}$ has four possible choices, depending on $R_1, R_2, q_{s_0}, q_{s_1}$.**

$q_{t_{n+1}} = \mathbb{P}(Y = 1 | T = t_{n+1}) = 0$ respectively. Using a similar analysis to the proof of Lemma 5.1, we can get that

$$\mathbb{P}(T = t_1 | S = s_n) = \mathbb{P}(Y = 1 | S = s_n) = q_{s_n},$$
$$\mathbb{P}(T = t_{n+1} | S = s_1) = \mathbb{P}(Y = 0 | S = s_1) = 1 - q_{s_1}. \tag{15}$$

From Lemma 4.1, we know that $\mathbb{P}(Y = 1 | S = s, T = t) \in \{0, 1\}$, $\forall s \in \mathcal{S}, t \in \mathcal{T}$. Denote set $\mathcal{T}_i = \{t_{i_k}\}_{k \in [K]}$, where $i \in \{2, \ldots, n\}$ and the value of $K$ is specified later, such that $\mathbb{P}(Y = 1 | S = s_{n+2-i}, T = t_{i_k}) = 0, \forall t_{i_k} \in \mathcal{T}_i$, and $\mathbb{P}(Y = 1 | S = s_{n+1-i}, T = t_{i_k}) = 1$. From Lemma 4.3, we know that the region $\mathcal{A}$ is in $\mathcal{T}$-upper-left, and therefore, we can get that $\forall i \in \{2, \ldots, n\}, t_{i_k} \in \mathcal{T}_i$,

$$\mathbb{P}(Y = 1 | S = s_j, T = t_{i_k}) = 1, \quad \text{iff } j \in [n + 1 - i]. \tag{16}$$

We set $K = 3n$ to ensure that $\{t_1, t_{n+1}, t_{i_k}\}_{i \in \{2, \ldots, n\}, k \in [K]}$ can capture any $\varepsilon$-inferentially-private Blackwell optimal information structure with at most $3n + 1$ output signals.

We then show that given a convex utility function $u$ and a joint distribution $P(S, Y)$, we can find an $\varepsilon$-inferentially-private information structure $P(Y, S, T)$ that maximizes the expected utility by a set of linear programs with different instantiations of $c_{i_k}^{(j)}$. Roughly, $c_{i_k}^{(j)}$ represents whether a secret-output pair $(s_j, t_{i_k})$ is associated with a wide cell with red outline ($c_{i_k}^{(j)} = 1$) or a narrow cell with blue outline ($c_{i_k}^{(j)} = e^\varepsilon$) in Fig. 4. From Lemma 4.3, we know that under a Blackwell optimal structure, the values of $c_{i_k}^{(j)}$ satisfy $\forall i \in \{2, \ldots, n\}, k \in [3n]$:

$$c_{i_k}^{(j)} \le c_{i'_{k'}}^{(j')}, \quad \forall j \in [n + 1 - i], \; j' \le j, \; i'_{k'} \preceq i_k,$$
$$c_{i_k}^{(j)} \le c_{i'_{k'}}^{(j')}, \quad \forall j \in [n] \setminus [n + 1 - i], \; j' \ge j, \; i'_{k'} \succeq i_k, \tag{17}$$

where $i'_{k'} \preceq i_k$ if $i' < i$ or $k' \le k$ when $i' = i$, and $i'_{k'} \succeq i_k$ if $i' > i$ or $k' \ge k$ when $i' = i$.

With the predefined values $c_{i_k}^{(j)} \in \{1, e^\varepsilon\}$, each program maximizes the expected utility $\mathbb{E}_t[u(q_t)]$ under the geometric characterizations of the $\varepsilon$-inferentially-private Blackwell optimal information structure.[3] The program is shown below. Specifically, conditions 19,20,23, and 24 calculate $p_T, q_T$ by $\mathbb{P}(S), \mathbb{P}(T|S)$ based on Eq. (16). Conditions 21 and 22 introduce the inferential privacy constraints for outputs $t_1$ and $t_{n+1}$ based on Eq. (15). Condition 25 calculates $l_{i_k}^{(j)} = \mathbb{P}(T = t_{i_k} | S = s_j)$, $\forall i \in \{2, \ldots, n\}, k \in [3n]$ based on the predefined value $c_{i_k}^{(j)} \in \{1, e^\varepsilon\}$. Conditions 26 and 27 introduce constraints on the value of $\mathbb{P}(T|S)$ based on Eq. (16). The variables in the program are $r_1^{(j)}$ ($j \in [n - 1]$), $r_{n+1}^{(j)}$ ($j \in \{2, \ldots, n\}$), and $l_{i_k}^{(n)}$ ($i \in \{2, \ldots, n\}, k \in [3n]$). Based on condition 25, we can rewrite condition 23 as $q_{t_{i_k}} = \frac{\sum_{j \in [n+1-i]} \mathbb{P}(S = s_j) \cdot c_{i_k}^{(j)}}{\sum_{j \in [n]} \mathbb{P}(S = s_j) \cdot c_{i_k}^{(j)}}$, which is fixed in each program. Therefore, our program is linear.

Finally, we provide a mechanism that maximizes the expected utility under utility function $u$ in Alg. 1. To design an optimal mechanism, we first enumerate all feasible $c_{i_k}^{(j)}$ based on constraints in Eq. (17). The number of enumerations is exponential with respect

---

[3]According to Thm. 2.1, the information structure that maximizes the expected utility under function $u$ must follow the geometric characterizations of the Blackwell optimal structure.

**ALGORITHM 1:** Information disclosure mechanism maximizing the expected utility for non-binary secrets.

---

**Input :** utility function $u$, distribution $\mathbb{P}(S, Y)$, inferential privacy level $\varepsilon$.

1 Enumerate $c_{i_k}^{(j)} \in \{1, e^{\varepsilon}\}, \forall i \in \{2, \ldots, n\}, j \in [n], k \in [3n]$, subject to the constraints in Eq. (17);

2 **for** each instantiation $c_{i_k}^{(j)}$: solve the optimization above;

3 $\hat{l}_1^{(j)}, \hat{l}_{n+1}^{(j)}, \hat{l}_{i_k}^{(j)} \leftarrow l_1^{(j)}, l_{n+1}^{(j)}, l_{i_k}^{(j)}$ that corresponds to the optimization achieving the maximal objective value;

4 Output the mechanism with

$$\mathbb{P}\left(T = t_1 | S = s_j, Y = 1\right) = \frac{\hat{l}_1^{(j)}}{q_{s_j}}, \quad \forall j \in [n],$$

$$\mathbb{P}\left(T = t_{i_k} | S = s_j, Y = 1\right) = \frac{\hat{l}_{i_k}^{(j)}}{q_{s_j}},$$

$$\forall j \in [n-1], \, i \in \{2, \ldots, n+1-j\}, \, k \in [3n],$$

$$\mathbb{P}\left(T = t_{n+1} | S = s_j, Y = 0\right) = \frac{\hat{l}_{n+1}^{(j)}}{1 - q_{s_j}}, \quad \forall j \in [n],$$

$$\mathbb{P}\left(T = t_{i_k} | S = s_j, Y = 0\right) = \frac{\hat{l}_{i_k}^{(j)}}{1 - q_{s_j}},$$

$$\forall j \in [n] \setminus \{1\}, \, i \in \{n+2-j, \ldots, n\}, \, k \in [3n].$$

---

to the number of secret $n$. The most straightforward way to obtain all feasible $c_{i_k}^{(j)}$ is to first enumerate the entire value space and then filter according to Eq. (17). Exploring more efficient enumeration methods may be an interesting direction for future work and could be of independent interest. For each enumeration, we solve the optimization described above. We can get an optimal structure—in the sense of maximizing expected utility under the utility function $u$—represented by $\hat{l}_1^{(j)}, \hat{l}_{n+1}^{(j)}, \hat{l}_{i_k}^{(j)}$, based on the optimization achieving the maximal expected utility. Finally, based on the fact that

$\mathbb{P}(S, Y, T) = \mathbb{P}(Y|S, T) \cdot \mathbb{P}(T|S) \cdot \mathbb{P}(S)$ and $\mathbb{P}(T|S, Y) = \frac{\mathbb{P}(S, Y, T)}{\mathbb{P}(S, Y)}$, we can design the optimal mechanism, represented by $\mathbb{P}(T|S, Y)$, that maximizes the expected utility under utility function $u$.

$$\max_{\substack{l_1^{(j)}, l_{n+1}^{(j)}, l_{i_k}^{(j)}, \\ \forall i \in \{2, \ldots, n\}, \, j \in [n], \\ k \in [3n]}} p_{t_1} \cdot u(q_{t_1}) + p_{t_{n+1}} \cdot u(q_{t_{n+1}}) + \sum_{\substack{i \in \{2, \ldots, n\} \\ k \in [3n]}} p_{t_{i_k}} \cdot u(q_{t_{i_k}}) \tag{18}$$

subject to $q_{t_1} = 1, \quad p_{t_1} = \sum_{j \in [n]} \mathbb{P}(S = s_j) \cdot l_1^{(j)}, \tag{19}$

$$q_{t_{n+1}} = 0, \quad p_{t_{n+1}} = \sum_{j \in [n]} \mathbb{P}(S = s_j) \cdot l_{n+1}^{(j)}, \tag{20}$$

$\forall j \in [n-1]:$

$$l_1^{(j)} = r_1^{(j)} \cdot l_1^{(n)}, \quad l_1^{(n)} = q_{s_n}, \quad r_1^{(j)} \in [e^{-\varepsilon}, e^{\varepsilon}], \tag{21}$$

$\forall j \in \{2, \ldots, n\}:$

$$l_{n+1}^{(j)} = r_{n+1}^{(j)} \cdot l_{n+1}^{(1)}, \quad l_{n+1}^{(1)} = 1 - q_{s_1}, \quad r_{n+1}^{(j)} \in [e^{-\varepsilon}, e^{\varepsilon}], \tag{22}$$

$\forall i \in \{2, \ldots, n\}, k \in [3n]:$

$$q_{t_{i_k}} = \frac{\sum_{j \in [n+1-i]} \mathbb{P}(S = s_j) \cdot l_{i_k}^{(j)}}{\sum_{j \in [n]} \mathbb{P}(S = s_j) \cdot l_{i_k}^{(j)}}, \tag{23}$$

$$p_{t_{i_k}} = \sum_{j \in [n]} \mathbb{P}(S = s_j) \cdot l_{i_k}^{(j)}, \tag{24}$$

$$l_{i_k}^{(j)} = \frac{c_{i_k}^{(j)}}{c_{i_k}^{(n)}} \cdot l_{i_k}^{(n)}, \quad l_{i_k}^{(n)} \geq 0, \quad \forall j \in [n-1], \tag{25}$$

$$\forall j \in [n]: \quad l_1^{(j)} + l_{n+1}^{(j)} + \sum_{\substack{i \in \{2, \ldots, n\} \\ k \in [3n]}} l_{i_k}^{(j)} = 1, \tag{26}$$

$$l_1^{(j)} + \sum_{\substack{i \in \{2, \ldots, n+1-j\} \\ k \in [3n]}} l_{i_k}^{(j)} = q_{s_j}. \tag{27}$$

