# OpenReview forum: "Inferentially-Private Private Information"
_ACM.org/TheWebConf/2025/Conference — WWW 2025 Poster_

### Official Review · Reviewer_MPBA · 2024-11-13

**Novelty:** 3
**Technical Quality:** 3

**Review:**

The paper considers an information disclosure problem that we need to satisfy inferential privacy constraints. However, the paper is quite hard to read as it lacks enough examples and intuitions, such as why sufficient conditions are hard to find. Also, most results rely on the assumption of binary secrets. It's important to provide some intuitions why general cases are intractable and the dependency on the cardinality of the secret space. Moreover, the paper lacks proof of results in sections 2 and 3. Even if most results are from other papers, showing why you can apply them directly under your setup is necessary. All in all, the problem considered is interesting, but some polishing seems necessary.

**Questions:**

1. For Thm 2.1, since $Q_1=Q_2+noise$, why does $E[u(Q_1)]\ge E[u(Q_2)]$ hold? Does it mean that the noise contains some information?

2. Many results look similar to the ones in [14]. What is the key technical innovation in your paper?

**Reviewer Confidence:**

2: The reviewer is willing to defend the evaluation, but it is likely that the reviewer did not understand parts of the paper

**Scope:**

3: The work is somewhat relevant to the Web and to the track, and is of narrow interest to a sub-community

---

### Official Review · Reviewer_63v4 · 2024-11-23

**Novelty:** 6
**Technical Quality:** 6

**Review:**

Summary: This paper studies an interesting extension of [He et al. 2022], where the perfect privacy constraint (following the notation in the paper) in [He et al. 2022] is relaxed to $\epsilon$-inferentially privacy. The paper mainly considers the two-state setup, in which a geometric characterization of Blackwell-optimal solutions is presented. For discrete settings with n secrets, the number of signals required for an optimal information structured is bounded. For the mechanism design problem with binary secrets and convex utility function, a closed-form solution is derived.

This paper is well-written and easy to follow.

I carefully read the main body of the paper (i.e., first 9 pages). Also, I went through in details the proofs of Lemma 4.1, Lemma 4.3, Theorem 4.1 and Lemma 5.1. I believe this paper is technically sound.

Strengths:
1.	Considering relaxing the perfect privacy to $\epsilon$-inferentially privacy is one important extension of [He et al. 2022], specifically, the Privacy-Preserving Recommendations problem in [He et al. 2022].
2.	While this paper mainly focuses on a binary-state setup, the results presented are indeed interesting. As shown in [He et al. 2022], characterizing the problems of general setups (e.g., more than two states) is challenging, which, I think, can be left as one future direction of this paper.
3.	I like the characterizations in Lemma 4.3. It is similar to the "upward-closed set" characterization in [He et al. 2022], but has one critical difference (please correct me if my understanding is wrong): the two regions $\mathcal{B}$ and $\mathcal{C}$ are ``separated’’ by some rectangles with width $L_t$ in this paper, while these two regions are immediately adjacent in [He et al. 2022].

Weaknesses:  Though I like some of the results in this paper (especially Lemma 4.3), my major concern is that the techniques used in the paper might be too specific, and at current stage, it is hard to see the potential of applying those techniques to other problems.  Also, while the results are interesting, they are not very surprising given that some similar characterizations exist in [He et al. 2022].

Overall I still believe it is an important extension of [He et al. 2022]. The strengths of this paper outweigh the weaknesses, and the results may contribute to the literature on privacy in information design.

**Questions:**

In the proof for Lemma 4.3, the intuitions for defining "H-L crossing blocks" is not quite clear. Can the authors discuss more about the intuitions and motivations for the definition of "H-L crossing blocks"?

**Reviewer Confidence:**

3: The reviewer is confident but not certain that the evaluation is correct

**Scope:**

3: The work is somewhat relevant to the Web and to the track, and is of narrow interest to a sub-community

---

### Official Review · Reviewer_WBbW · 2024-12-03

**Novelty:** 5
**Technical Quality:** 7

**Review:**

This paper considers a constrained information design problem. An information designer must disclose as much information as possible about a relevant state variable Y. At the same time, Y is also correlated with a secret variable S which is also known to the designer. The designer must therefore design a Blackwell experiment, taking (Y,S) and outputting a distribution over signals which in turn induces a posterior. The goal is to produce a Blackwell-optimal information structure with respect to Y, subject to the constraint that the beliefs about S do not update by more than a fixed multiplicative factor ("inferential privacy"). This is an extension of existing work, which considered the same problem subject to exact inferential privacy (no updates to the posterior over S).

The authors give structural results that hold broadly for this problem, showing that many observations from existing work on exact privacy continue to hold in the setting with the newly relaxed privacy constraint. They give an exact characterization of the unique optimal solution in the case where the secret is binary, which ends up being relatively clean, and a bound on the number of signals required for the general case.

Main strengths:
 • Very well written. I was able to follow the technical content, in no small part because of the many very helpful figures.
 • The problem is simple, natural, and feels fundamental.
 • The binary case, while a bit limited, is not so limited as to be uninteresting. There is a long history of information design papers starting out in the binary setting.
 • The bound on the number of signals was also quite interesting.

Weaknesses:
 • Despite the claims that the techniques are new, I have a bit of trouble believing that this is true in a sufficiently strong sense. It seems like most of the arguments are still variants of greedy exchange arguments or arguments about necessary conditions for local optimality. ("If condition X does not hold, here's an easy improvement.")

**Questions:**

The general bound on the number of signals (3|S| + 1) is larger than the binary bound (4). Why is this?
I'm curious about computational complexity of the finding optimal information structures with general secrets. Is anything known? (And am I correct that the algorithm given in the present paper is exponential-time?)

**Reviewer Confidence:**

3: The reviewer is confident but not certain that the evaluation is correct

**Scope:**

3: The work is somewhat relevant to the Web and to the track, and is of narrow interest to a sub-community